# Size matters: Large copy number losses in Hirschsprung disease patients reveal genes involved in enteric nervous system development

**Laura E. Kuil**[1], **Katherine C. MacKenzie**[1], **Clara S. Tang**[2,3], **Jonathan D. Windster**[1,4], **Thuy Linh Le**[5], **Anwarul Karim**[2], **Bianca M. de Graaf**[1], **Robert van der Helm**[1], **Yolande van Bever**[1], **Cornelius E. J. Sloots**[4], **Conny Meeussen**[4], **Dick Tibboel**[4], **Annelies de Klein**[1], **René M. H. Wijnen**[4], **Jeanne Amiel**[5], **Stanislas Lyonnet**[5], **Maria-Mercè Garcia-Barcelo**[2], **Paul K. H. Tam**[2,3], **Maria M. Alves**[1], **Alice S. Brooks**[1], **Robert M. W. Hofstra**[1,6], **Erwin Brosens**[1]*

1 Department of Clinical Genetics, Erasmus MC–Sophia Children's Hospital, University Medical Center Rotterdam, Rotterdam, The Netherlands, 2 Department of Surgery, Li Ka Shing Faculty of Medicine, The University of Hong Kong, Hong Kong, China, 3 Li Dak-Sum Research Centre, The University of Hong Kong–Karolinska Institutet Collaboration in Regenerative Medicine, Hong Kong, China, 4 Department of Paediatric Surgery, Erasmus MC–Sophia Children's Hospital, University Medical Center Rotterdam, Rotterdam, The Netherlands, 5 Laboratory of embryology and genetics of malformations, Institut Imagine Université de Paris INSERM UMR1163 Necker Enfants malades University Hospital, Paris, France, 6 Stem Cells and Regenerative Medicine, UCL Great Ormond Street Institute of Child Health, London, United Kingdom

* e.brosens@erasmusmc.nl

**Data Availability Statement:** All relevant data are within the manuscript and its Supporting information files. CNV regions are submitted to

## Abstract

Hirschsprung disease (HSCR) is a complex genetic disease characterized by absence of ganglia in the intestine. HSCR etiology can be explained by a unique combination of genetic alterations: rare coding variants, predisposing haplotypes and Copy Number Variation (CNV). Approximately 18% of patients have additional anatomical malformations or neurological symptoms (HSCR-AAM). Pinpointing the responsible culprits within a CNV is challenging as often many genes are affected. Therefore, we selected candidate genes based on gene enrichment strategies using mouse enteric nervous system transcriptomes and constraint metrics. Next, we used a zebrafish model to investigate whether loss of these genes affects enteric neuron development *in vivo*.

This study included three groups of patients, two groups without coding variants in disease associated genes: HSCR-AAM and HSCR patients without associated anomalies (HSCR-isolated). The third group consisted of all HSCR patients in which a confirmed pathogenic rare coding variant was identified. We compared these patient groups to unaffected controls. Predisposing haplotypes were determined, confirming that every HSCR subgroup had increased contributions of predisposing haplotypes, but their contribution was highest in isolated HSCR patients without *RET* coding variants. CNV profiling proved that specifically HSCR-AAM patients had larger Copy Number (CN) losses. Gene enrichment strategies using mouse enteric nervous system transcriptomes and constraint metrics were used to determine plausible candidate genes located within CN losses. Validation in zebrafish using CRISPR/Cas9 targeting confirmed the contribution of *UFD1L*, *TBX2*, *SLC8A1*, and *MAPK8*

ClinVar, ID's are included in the manuscript
(Table 1). Our ethics committee does not allow
sharing of individual patient or control genotype
information in the public domain. This study makes
use of data generated by the DECIPHER
community. A full list of centers who contributed to
the generation of the data is available from http://
decipher.sanger.ac.uk and via email from
decipher@sanger.ac.uk.

**Funding:** PT is funded by the Theme-based
Research scheme [grant number T12C-714/14-R]
and CT by the Health Medical Research Fund
[HMRF 06171636]. RH was funded by a grant
from the Sophia Foundation [grant nr. S14-33].
The salary of KM was paid by S14-33. The funders
had no role in study design, data collection and
analysis, decision to publish, or preparation of the
manuscript.

**Competing interests:** The authors have declared
that no competing interests exist.

to ENS development. In addition, we revealed epistasis between reduced Ret and Gnl1 expression and between reduced Ret and Tubb5 expression *in vivo*. Rare large CN losses—often *de novo*—contribute to HSCR in HSCR-AAM patients. We proved the involvement of six genes in enteric nervous system development and Hirschsprung disease.

## Author summary

Hirschsprung disease is a congenital disorder characterized by the absence of intestinal neurons in the distal part of the intestine. It is a complex genetic disorder in which multiple variations in our genome combined, result in disease. One of these variations are Copy Number Variations (CNVs): large segments of our genome that are duplicated or deleted. Patients often have Hirschsprung disease without other symptoms. However, a proportion of patients has additional associated anatomical malformations and neurological symptoms. We found that CNVs, present in patients with associated anomalies, are more often larger compared to unaffected controls or Hirschsprung patients without other symptoms. Furthermore, Copy Number (CN) losses are enriched for constrained coding regions (CCR; genes usually not impacted by genomic alterations in unaffected controls) of which the expression is higher in the developing intestinal neurons compared to the intestine. We modelled loss of these candidate genes in zebrafish by disrupting the zebrafish orthologues by genome editing. For several genes this resulted in changes in intestinal neuron development, reminiscent of HSCR observed in patients. The results presented here highlight the importance of Copy Number profiling, zebrafish validation and evaluating all CCR expressed in developing intestinal neurons during diagnostic evaluation.

## Introduction

The enteric neurons and glia in the intestine form the enteric nervous system (ENS), which is derived from the neural crest. Enteric neural crest cells (ENCCs) invade the developing gut tube in the proximal foregut around week four of human development and migrate caudally to colonize the distal hindgut in week 7 [1]. Failure in either migration, proliferation, differentiation or survival of these ENCCs [2] is known to result in Hirschsprung disease (HSCR). This congenital enteric neuropathy is characterized by an absence of the ENS, or also called enteric ganglia, in the distal colon (called aganglionosis). The length of the affected region is most often restricted to the recto-sigmoid region, however the length of aganglionosis can extend to the whole intestine. HSCR can segregate through families [3–6] and, depending on the length of the affected region and gender of the proband, siblings can have a recurrence risk of up to 33% [7]. Most patients have HSCR as an isolated anatomical malformation [8] and in these patients this disease is a textbook example of a complex genetic disorder where the phenotype stems from a combination of genetic alterations: rare coding variants, predisposing common haplotypes and losses or gains of (large) parts of the genome (Copy Number Variation (CNV) [9]. The common haplotypes influence HSCR penetrance and include haplotypes near *RET*. They are known to increase disease risk substantially, especially if homozygous, in specific combinations (rs2506030, rs7069590, rs2435357) [10,11], together with other risk loci near the

semaphorin gene cluster (rs80227144) [11,12], or the neuregulin 1 gene (*NRG1*; rs7005606) [13–15].

Approximately 18% of HSCR patients have additional anatomical malformations or neurological symptoms (HSCR-AAM) [7]. The combination of disease entities found in HSCR-AAM patients could be the result of one or more point mutation(s) or inherited variants. However, CNVs have been found to contribute to the etiology of many traits, common diseases and congenital anomalies [16]. In addition, chromosomal deletions have proven instrumental in the identification of disease-causing genes. For example, deletions of 10q11 [17,18] led to the identification of the REarranged during Transfection gene (*RET)*, the major responsible gene for familial and sporadic isolated HSCR [19,20]. Also for the identification of causal genes for several syndromes with HSCR as a key feature. For example, deletions of 13q [9, 21–25] resulted in the identification of one of the genes responsible for Waardenburg-Shah syndrome type 4 (*EDNRB*) [26]. Deletions of 2q [27–30] and 4p [31] contributed to the discovery of genes responsible for Mowatt-Wilson syndrome (*ZEB2*, formerly *ZFHX1B*) [32] and the gene responsible for Congenital Central Hypoventilation Syndrome *(PHOX2B)* [33]. Therefore, we believe that rare CNVs could significantly contribute to patients with HSCR, in particular to HSCR-AAM patients.

Large CNVs harbour many genes making it difficult to determine genes responsible for the observed trait or disease. One of the strategies to identify the genes involved in disease development is to disrupt these genes in animal models. As an *in vivo* model to study genetic alterations that affect the development of the enteric nervous system we use zebrafish (*Danio rerio*), which are often used in development biology as they are easily accessible due to their rapid and ex-utero development. By injecting CRISPR/Cas9 complexes in the one cell stage of fertilized oocytes one can relatively easily generate various insertions and deletions that will disrupt gene function. This method has such high efficiency that it enables us to study the effects of gene disruption in the injected F0 generation zebrafish larvae [34–36]. In addition, the development of the zebrafish enteric nervous system is highly similar to that in mouse and human [37,38]. Therefore, the zebrafish makes an excellent model to use for functional genetic testing of HSCR candidate genes *in vivo*.

In this manuscript, we test our hypothesis that rare CNVs could significantly contribute to patients with HSCR, in particular to HSCR-AAM patients. Hence, we compared the CNVs of HSCR patients with and patients without associated anomalies, which lack rare pathogenic coding changes, and HSCR patients with a known pathogenic coding mutation or variant. Our data pointed to the presence of larger copy number (CN) losses in HSCR-AAM patients. Subsequently, we aimed to identify the gene(s) in these CNV(s) contributing to ENS development by disruption of their orthologues in zebrafish in order to determine whether their loss affects ENS development.

## Results

We selected 58 out of 197 HSCR patients for which DNA and informed consent were available, and in whom at least the *RET* gene was screened. HSCR patients were subdivided in three groups for the CNV detection study: patients with HSCR and additional anatomical malformations or neurological defects (HSCR-AAM), but without a *RET* pathogenic variant, or other causal genetic defect (group 1, n = 23, Table 1 and Fig 1A), patients with HSCR and a known variant in *RET* or another causal gene (group 2, n = 15, Table 2 and Fig 1A), and patients with only HSCR, without a deleterious *RET* coding variant or other causal genetic defect (group 3, n = 20, Fig 1A). Additionally, we included unaffected control individuals (group 4, n = 326, Fig 1A).

**Table 1. Hirschsprung patients without a *RET* mutation and additional phenotypical features.**

| Patient | HSCR type | Other phenotypical characteristics |
|---|---|---|
| P_000482 | Short | Hydrocephalus, macrocephaly, autism |
| P_000540 | Short | Facial dysmorphisms |
| P_000494 | Short | Cardiac defects (VSD, ASD PDA, tricuspid atresia), dysplastic ears, renal malrotation |
| P_000512 | Short | Epilepsy, intellectual disability |
| P_000553 | Short | Cardiac defects (VSD, dextrocardia, PDA, double outlet right ventricle), intestinal malrotation |
| P_000559 | Total colonic | Dysmorphic features, tracheomalacia, cardiac defects (dilated left ventricle, absence of AV conduction) |
| P_000561 | Short | Facial dysmorphisms, small fontanelle, gastro-esophageal reflux, laryngeal web |
| P_000555 | Short | Hypoplastic thumb, hearing loss, developmental delay, facial dysmorphisms |
| P_002459 | Short | Hypospadias, mild autism |
| P_000567 | Short | Facial dysmorphisms, hearing loss, microcephaly, immunological hypersensitivity, nevus flammeus |
| P_000536 | Abnormal | Telecanthus upslant; short segment HSCR although a longer segment is abnormal ganglionated. |
| P_000562 | Short | Cafe au lait spots, cardiac defect (VSD) |
| P_000572 | TIA | Retrognathia, skin abnormality, facial dysmorphisms, cardiac defect (pulmonary valve stenosis) |
| P_000568 | Short | Dysmorphic features, hydrocele testis, hemangioma |
| P_000478 | Short | Hypertelorism, facial dysmorphisms |
| P_000520 | Short | Mild facial dysmorphisms, sandal-gap of toe |
| P_002455 | Short | Hypermobility of fingers; mild developmental delay, downslant of eyes |
| P_001763 | Short | White hair lock, mild developmental day |
| P_000537 | Short | Gross motor delay, spastic hemiplegia, bronchopulmonary dysplasia, cardiac defect (PDA) |
| P_000528 | Total colonic | Intellectual disability |
| P_000573 | Short | Epicanthal folds, small ears, broad eyebrows with mild synophrys |
| P_002450 | Long | Developmental delay |
| P_002343 | Short | Hypertelorism, long deeply grooved philtrum |

Depicted are the patients with HSCR and additional anatomical malformations or neurological defects (HSCR-AAM), but without a *RET* pathogenic variant, or other causal genetic defect (group 1, n = 23). Listed are the patient identifiers, HSCR classification, and phenotypical description regarding the associated anomalies.

## Rare large CNVs are enriched in HSCR-AAM patients

Patient and control CNVs were determined and classified as a "rare CNV" when absent from large control cohorts (n = 19,584). We also included known pathogenic or modifier CNVs [39,40]. The size, type and gene content characteristics of rare CNVs were compared to those of 326 unaffected controls (Group 4, n = 326). With this approach, 56 rare CNVs were detected in 34 HSCR patients (S1 Table). In group 1, 10 CN losses, 8 CN gains and 1 maternally inherited hemizygous loss on chromosome X in a male patient, were detected. In group 2, 5 CN losses and 8 CN gains were identified, and in group 3, 7 CN losses, 12 CN gains, 4 homozygous losses and 1 hemizygous loss on chromosome X in a male patient, were found. None of the CNVs present in any of our patients affected known HSCR genes (S1 and S3 Tables).

We determined segregation of rare CNVs in nine patients, five of these were *de novo*: three CN gains (22q11.21—q11.22, 22q11.21 and 7q36.1) and two losses (17q23.1—q23.2 and 6p22.1—p21.33. Two CNVs were inherited maternally (10q11.22—q11.23 loss and Xq28 loss).

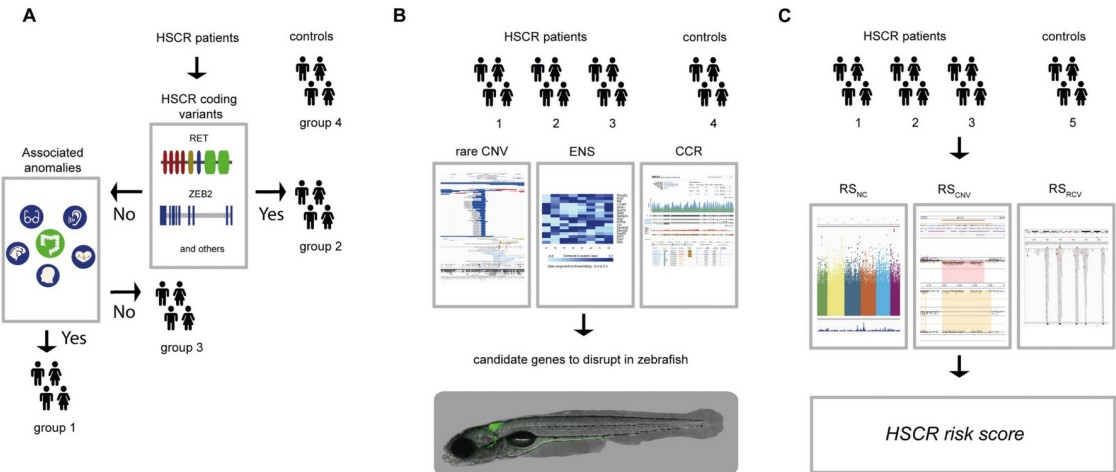

**Fig 1. Schematic overview of our overall study design and methods used.** (A) For this study we included 326 controls (group 4) and 58 HSCR patients. We determined the RET and / or known disease gene coding mutations of HSCR patients (n = 58). The HSCR patients with coding mutations were included in group 2 (n = 15). We determined the presence of associated anomalies in the rest of the HSCR patients, including them in either the group containing associated anomalies (HSCR-AAM; group 1, n = 23) or in the group without associated anomalies (group 3, n = 30). (B) For all subgroups of HSCR patients and the controls Copy Number profiles were determined. To select candidate genes, we ranked CNVs according to their frequency in the unaffected controls from the Deciphering Developmental Disorders project (for illustrative purposes a screenshot of their UCSC genome browser track; https://genome-euro.ucsc.edu/ is depicted). We determined if genes included in the rare CNVs were "ENS genes": genes with increased expression in isolated ENS (with and without the addition of GDNF) cells compared to the whole intestine (for illustrative purposes a screenshot of a heatmap of differential gene expression of known disease genes derived from brb-array tools (https://brb.nci.nih.gov/BRB-ArrayTools/) is depicted. Additionally, we determined if a gene was known as a Constrained Coding Region (CCR) (For illustrative purposes a screenshot of gnomAD browser website where such scores can be found; https://gnomad.broadinstitute.org/). Next, we determined whether disruption of the main candidate genes resulted in a reduction of enteric neurons in zebrafish. (C) In parallel, we evaluated the contribution of predisposing haplotypes across groups. Combined, we suggest that the RSnc (predisposing haplotypes), RSrcv (deleterious rare variant burden) and RScnv (deleterious Copy Number Variation) result in the genetic risk for Hirschsprung disease. Abbreviations: CNV = Copy Number Variation, ENS = Enteric Nervous System, RSnc = Risk Score non-coding variants, RSrcv = Risk Score rare coding variants, RScnv = Risk Score CNV, CCR = Constrained Coding Region.

Two of the homozygous losses in patient P_000490 (group 3), disrupt the Cystic fibrosis transmembrane conductance regulator (CFTR) locus. However, this patient does not show signs of cystic fibrosis. The inheritance pattern of other rare CNVs could not be determined due to unavailability of parental DNA.

The number of rare CNVs per individual did not differ between subgroups or controls (p = 0.385, Fig 2A). However, CNVs found in HSCR-AAM patients (Group 1) were larger compared to controls (Group 4, p = 7.297E-6, Fig 2B and S2 Table). This difference is attributed to outliers, large size of losses in specific patients of group 1 (p = 3.15E-08)(Fig 2C and S2 Table and S1 Fig), strongly suggesting a role of these specific CNVs in these HSCR-AAM patients.

## Candidate genes within a CN loss are constrained coding regions (CCR) and expressed in the ENS

HSCR disease genes (S3 Table) are often constrained coding regions (CCR; genes usually not impacted by genomic alterations in unaffected controls) and expressed in the affected tissue: the ENS. We hypothesize that genes within a CNV region contributing to HSCR should share these characteristics (Fig 1B and S3 Table). Therefore, we used a previously published microarray dataset and considered genes as "ENS genes" if the gene expression is higher in isolated "ENS cells" (with and without the addition of GDNF) compared to the expression in other

**Table 2. HSCR patients with a deleterious variant.**

| Patient | HSCR type | Other phenotypical characteristics | Genetic defect |
|---|---|---|---|
| P_000302 | Short | No | NM_002181.3 (IHH):c.151C>A, NM_000168.5 (GLI3): c.2119C>T, NM_001190468.1 (GDNF):c.676_681delGGATGT |
| P_000526 | Short | No | NM_020630 (RET):c.1196C>T |
| P_000479 | Long | No | NM_020630 (RET):c.656-21C>T |
| P_000502 | Short | No | NM_020630 (RET):c.1880_1892del |
| P_002442 | Long | No | NM_020630 (RET):c.2599C>T |
| P_000566 | Short | No | NM_020630 (RET):c.3173A>G |
| P_000544 | Long | No | NM_020630 (RET):c.2690G>A |
| P_000534 | Short | Hypospadias, anorectal malformation type perineal fistula | NM_020630 (RET):c.2906G>A |
| P_000480 | Short | Short stature | NM_013956 (NRG1):c.811A>T |
| P_004502 | Short | Hypertelorism, triangular face, pointy chin, straight eye brows, deepset eyes, small dysmorphic ears, agenesis of corpus callosum, hypospadia, dysmorphic nose | NM_014795 (ZEB2): c.1570del |
| P_000557 | Total colonic | Postaxial polydactyly | NM_020630 (RET):c.C229C>T |
| P_000518 | Short | Facial dysmorphisms, microcephaly, bilateral generalized polymicogyria, developmental delay, short stature, hypotonia, eye anomaly | NM_015634 (KIFBP):c.268C>T |
| P_000486 | Total colonic | No abnormal phenotypedescribed. Normal psychomotor development | NM_001122659 (EDNRB):c.534_535insGGTGCCT |
| P_000570 | Short | Congenital central hypoventilation syndrome | NM_003924 (PHOX2B):c.738_761dup |
| P_000576 | Short | Microcephaly, epicantus folds, upslant eyes, broad nose, synophrys naevus sacralis hyperpigmentosis back and shoulders, abnormal palmar creases | NM_020630 (RET):c.1321A>C,: NM_020630 (RET):c. C1941C>T |

Depicted are the patients with HSCR and a known variant in *RET* or another causal gene (group 2, n = 15) Listed are the patient identifiers, HSCR classification, a short phenotypical description regarding associated anomalies if applicable, and the genetic defect.

intestinal cells or total intestines in mice [41,42]. Rare CNV impacted a gene or transcript (e.g. microRNA, long noncoding RNA) 1502 times (1213 unique ID's) in groups 1–4 (S12 Table). We were not able to evaluate the expression of 519 genes or transcripts. Of these, 499 did not have a known mouse orthologue. Often, these genes or transcripts lacking a mouse orthologue were microRNA, long noncoding RNA or were genes of which the symbol is not available. Twenty-two genes or transcripts did not have probes in the microarray datasets used. In total the rare CNVs included 281 "ENS genes" (n = 91 in group 1–3, n = 190 in group 4) [41,42]. All rare CNVs containing "ENS genes" are depicted in S4 and S12 Tables. Compared to controls, the rare CNVs in group 1 were significantly enriched for "ENS genes" (p = 4,93E-08, Fig 2D and S2 Table), in particular in the CN losses (p = 3,2604E-10, Fig 2E and S2 Table). In addition, CCR were enriched in rare CNVs of group 1 compared to controls (p = 4,63E-05, Fig 2F and S2 Table), again in particular in rare CN losses (p = 8,13E-06, Fig 2G and S2 Table). Moreover, losses in HSCR-AAM patients (Group 1) contained more "ENS genes" that are a CCR [43,44], compared to losses found in controls (Group 4, p = 2,20E-04, Fig 2H). Hence, "ENS genes" that are also a CCR within CN losses were prioritized as candidate genes for HSCR and included in zebrafish evaluation studies: *AKT3, GNL1, GABBR1, SLC8A1, MAPK8, UFD1L, TBX2, USP32* and *TUBB* (Table 3).

## HSCR disease coding genetics is heterogeneous

Stronger evidence for the involvement of these CNVs or genes included in CNVs could be provided by their involvement in additional Hirschsprung disease patients. First, we evaluated if

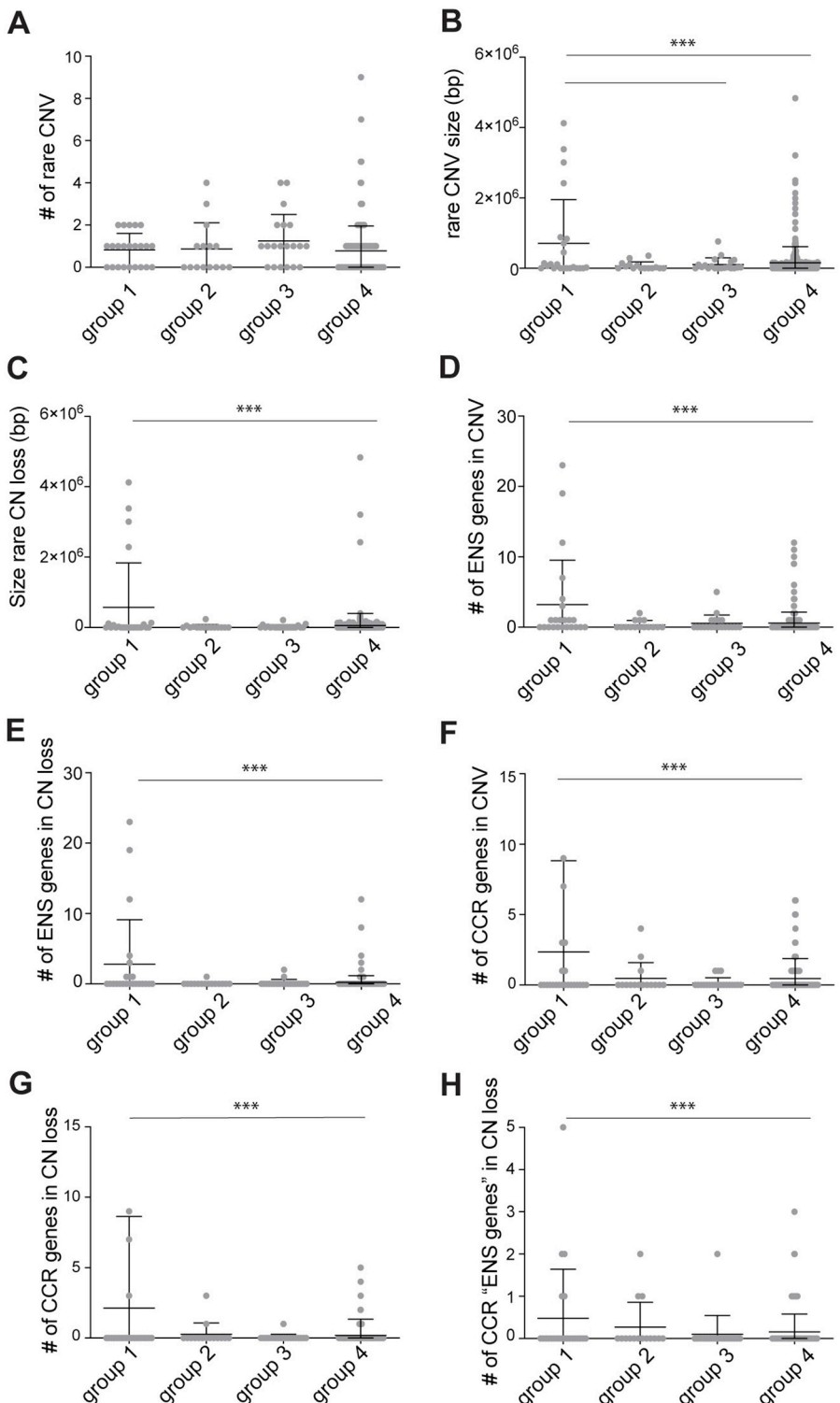

**Fig 2. Copy Number Variation analysis of HSCR patients and controls.** (A) Graph showing the number of rare CNVs found in the four groups. No statistical differences were found. (B) Graph showing the size in base pairs (bp) of the rare CNVs found in the four groups. Rare CNV size in group 1 (HSCR-AAM) was significantly larger than those found in group 3 (HSCR without a causal variant identified) and group 4 (controls). (C) Graph showing the size (bp) of the rare CN losses found in the four groups. Rare CN loss size in group 1 (HSCR-AAM) was significantly larger than those found in group 4 (controls). (D) Graph showing the number of "ENS genes" included in the CNVs found in the

four groups. The number of "ENS genes" included in CNVs of group 1 (HSCR-AAM) was significantly higher than those found in group 4 (controls). (E) Graph showing the number of "ENS genes" included in the CN losses found in the four groups. The number of "ENS genes" included in CN losses of group 1 (HSCR-AAM) was significantly higher than those found in group 4 (controls). (F) Graph showing the number of CCR genes included in the CNVs found in the four groups. The number of CCR genes included in CNVs of group 1 (HSCR-AAM) was significantly higher than those found in group 4 (controls). (G) Graph showing the number of CCR genes included in the CN losses found in the four groups. The number of CCR genes included in CN losses of group 1 (HSCR-AAM) was significantly higher than those found in group 4 (controls). (H) Graph showing the number of "ENS genes" that are also a CCR, included in the CN losses found in the four groups. These genes were considered as the HSCR candidate genes. The number of CCR "ENS genes" included in CNVs of group 1 (HSCR-AAM) was significantly higher than those found in group 4 (controls). Each dot represents one patient. Error bars represent standard deviation. Statistical analysis used: one-way ANOVA followed by students t-test. ** $p < 0.01$ *** $p < 0.001$; Exact p-values and statistics can be found in S2 Table.

there was CNV overlap within this cohort of patients. Within our small cohort only two CNV regions overlapped: (1) a gain in patients P_000544, P_000566 (group 2) and P_001636 (group 3), covering the blood group type gene *RHCE* (1p36.11), and (2) a loss in patients P_000557 (group 2) and P_000515 (group 3) of which only the *ETFDH* gene is affected in both patients (4q32.1). However, *RHCE* and *ETFDH* are no "ENS genes" [41,42].

Next, we evaluated if there was CNV overlap between our cohort of patients and those of previously published HSCR patients. The 22q11 deletion described in one of the patients included in DECIPHER (ID 249397) had overlap with the 22q11 deletion found in patient P_000561 (chr22:21032298–21630630). Although several genes are located in this region, only *LZTR1* is an "ENS gene" [41,42]. However, this is not a CCR. One of the losses described in another DECIPHER patient (ID 249405) overlaps with the 1q44 loss present in patient P_002431, with *AKT3* as the only gene affected by both CNVs. This is an CCR and an "ENS gene" [41,42] and was thus included as a candidate gene in this study. Comparing our data to that of a previously described HSCR cohort [45], only five genes in a CN loss in our cohort were also affected in that dataset. However, none of these five genes were CCR.

In an independent whole exome sequencing cohort of patients with HSCR, we determined the presence of putative deleterious alterations in all genes affected by a rare CNV in our cohort. Nonsense or splice-site variants were detected in eight genes (Table 4). Only one variant, detected in DNA derived from blood and isolated ENS cells, affected an "ENS gene" that is also CCR and present in a CN loss: a frameshift variant (*TUBB*; NM_001293212.1: c.1330_1331delCAinsA, p.Gln444Serfs*35) in a patient with isolated HSCR (Table 3).

**Table 3. "ENS genes" that are a CCR in rare CN losses and that have a zebrafish orthologue.**

| Patient | RS$_{NC}$ | Gene | ExAC/ GnomAD intolerance scores | | | | CNV region |
| | | | Missense Z | pLI | Deletion single | Deletion CNV | |
|---|---|---|---|---|---|---|---|
| P_000479 | 4,16 | *SLC8A1* | 2.23 | 1.00 | -0.02 | 0.31 | chr2:40,624,267–40,646,501 |
| P_000512 | 8,31 | *TUBB*# | 5.71 | 0.98 | -2.85 | -1.72 | chr6:28,005,012–31,683,185 |
| | | *GNL1* | 2.52 | 1.00 | 1.03 | 0.70 | |
| | | *GABBR1* | 4.98 | 1.00 | 1.36 | 1.23 | |
| P_000537 | 6,92 | *MAPK8* | 2.92 | 1.00 | 0.84 | -2.25 | chr10: 49,033,586–52,431,193 |
| P_000561 | 6,08 | *UFD1L* | 2.77 | 1.00 | 1.06 | -2.53 | chr22:18,861,209–21,630,630 |
| P_000567 | 7,98 | *TBX2* | 1.75 | 0.99 | -0.01 | 0.53 | chr17:58,076,721–60,362,868 |
| | | *USP32* | 3.55 | 1.00 | -2.85 | -0.93 | |
| P_002431 | 9,37 | *AKT3* | 4.03 | 1.00 | -2.61 | -1.20 | chr1:243,963,527–244,016,804 |

Genes marked with an # also have a loss of function variant in an independent HSCR cohort (see Table 4). Depicted are the risk score non coding (RSnc)(see S7 Table) and the CNV and variant intolerance scores (missense Z/pLI) derived from http://gnomad.broadinstitute.org/ and http://exac.broadinstitute.org/about.

**Table 4. WES: Nonsense and splice site variants in rare CNV genes in HSCR patients.**

| Sample | Chr | Start | Stop | Ref | Alt | Exon | Gene | Type | location | Effect | HGVS cDNA-level | CADD | gnomADv2.1 Exomes | gnomADV2.1 Genomes |
|---|---|---|---|---|---|---|---|---|---|---|---|---|---|---|
| SE14-0527 | 17 | 59946466 | 59946466 | | T | 23 | INTS2 | insertion | exonic | frameshift | NM_001330417.1:c.3172dupA | | 0 | 0 |
| HK19-0006 | 17 | 60072727 | 60072727 | C | T | 10 | MED13 | snv | intronic | splicing | NM_005121:c.1968-1G>A | 22.4 | 0 | 0 |
| SE16-3114 | 17 | 59469360 | 59469360 | C | T | 26 | BCAS3 | snv | exonic | stopgain | NM_001320470.1:c.2773C>T | 18.4 | 0.0002 | 0.0007 |
| SE14-0656 | 6 | 31604286 | 31604286 | G | T | 27 | PRRC2A | snv | splicing | splicing | NM_004638.3:c.5836-1G>T | 23.6 | 0.0001 | 3.24E-05 |
| SE16-3123 | 6 | 30692109 | 30692110 | CA | A | 4 | TUBB | substitution | exonic | frameshift | NM_001293212.1: c.1330_1331delCAinsA | . | 0 | 0 |
| SE17-3220 | 7 | 151962296 | 151962296 | T | C | 8 | KMT2C | snv | splicing | splicing | NM_170606.2:c.1013-2A>G | 22 | 8.21E-06 | 0 |
| HK19-0002 | 9 | 28476025 | 28476025 | T | G | | LINGO2 | snv | intronic | splicing | NM_001258282:c.-395-2A>C | 23.2 | 0 | 0 |
| HK19-0003 | 10 | 52103343 | 52103344 | TA | T | 7 | SGMS1 | deletion | exonic | frameshift | NM_147156:c.T529delT-:p. F177del | 33 | 0 | 0 |
| HK19-0004 | 10 | 52104106 | 52104108 | CTG | C | | SGMS1 | deletion | intronic | splicing | NM_147156:c.-313-2CAG>—G | 24.5 | 0 | 0 |
| HK19-0005 | 10 | 52349911 | 52349911 | A | G | | SGMS1 | snv | intronic | splicing | NM_147156:c.-683+2T>C | 23 | 0 | 0 |
| SE16-3109 | 22 | 21065731 | 21065731 | G | A | 51 | PI4KA | snv | exonic | stopgain | NM_058004.3:c.5821C>T | 51 | 0.0002 | 0.0002 |
| HK19-0001 | 3 | 14485130 | 14485130 | A | G | | SLC6A6 | snv | intronic | splicing | NM_001134367:c.297-6A>G | 15.92 | 0 | 0 |

Rare putative deleterious nonsense variants and variants predicted to affect splicing in a whole exome sequencing cohort of HSCR (n = 76, 149 controls) [95] and whole genome sequencing cohort of 443 short segment HSCR patients and 493 unaffected controls [46]. Variants in genes intolerant to variation that were also impacted by the de novo 17q23.1—q23.2 loss (INTS2, MED13), the de novo 6p22.1—p21.33 loss (PRRC2A, TUBB), the 9p21 loss (LINGO2), the maternal inherited 10q11.22—q11.23 loss (SGMS1), de novo 7q36.1 gain (KMT2C) and the 3q24 gain (SLC6A6). TUBB is highlighted in gray as this gene is included in the candidate list.

However, there was no significant enrichment in HSCR patients for nonsense, splice-site or missense variants in genes impacted by rare CN gains nor CN losses (S5 Table) when comparing 443 short segment HSCR patients and 493 controls [46], likely underlining the genetic heterogeneity between HSCR patients.

Thus, for two of our HSCR candidate genes we identified an additional HSCR patient: one containing a CN loss including *AKT3*, and a patient carrying a frameshift variant in *TUBB*.

## Zebrafish validation confirms the impact of candidate gene disruption on ENS development

To validate the effect of losing one copy of these candidate genes on ENS development, we disrupted their orthologues in zebrafish (*akt3a*, *akt3b*, *gnl1*, *gabbr1a*, *gabbr1b*, *slc8a1a*, *slc8a1b*, *mapk8a*, *mapk8b*, *ufd1l*, *tbx2a*, *tbx2b*, *usp32* and *tubb5*) by CRISPR/Cas9 targeting (Fig 3A). Instances in which there are two variants (a and b) of the gene in zebrafish, both variants were targeted simultaneously using two guide RNAs (gRNAs). To determine whether the larvae present an ENS phenotype five categories were made (Fig 3B) [38]. Larvae were categorized as category I when full colonization was observed with a regular distribution of the ENS along the length and width of the intestine by evaluating the pattern of GFP+ cells in the tg(*phox2bb*: GFP) transgenic reporter line [47]. This reporter line labels crest-derived neurons and progenitors in the intestine. Larvae containing an overall obvious reduction in the number of ENS cells were classified as category II. In case only the most distal end of the intestine lacks their ENS this was classified as category III (Fig 3B, indicated with the arrowhead). In case the migration of the ENS halted somewhere in the mid-gut of the zebrafish, as depicted by the green arrowheads in Fig 3B, fish were classified as category IV. Most severe cases, where total absence of an ENS in the mid-gut and distal-gut was observed were categorized as category V (Fig 3B). For statistical analysis the percentages of fish presenting with a category I, or in other words an unaffected ENS, was used (white bars in the graph of Fig 3C). Premature termination of ENS migration or reduced presence of the ENS was observed in a subgroup of larvae injected with gRNAs targeting *ufd1l* (p = 0,0166) and *slc8a1a/b* (p = 0,0073). Several of the fish in which we disrupted *tbx2a/b* (p = 0,0593), or *mapk8a/b* (p = 0,0541, Fig 3A, 3B and 3C) presented with phenotypes in category II, III and IV or category II and III respectively, but did not reach significance. Quantification of the number of ENS cells normalized to the intestinal length showed that also in overall numbers disruption of *ufd1l* (p = 0,0235) and *slc8a1a/b* (p = 0,0013) had a significant effect (Fig 3D). In addition, the effect of *mapk8a/b* disruption was also significant (p = 0,0254, Fig 3D). To confirm our findings, we repeated the experiments disrupting *tbx2a/b* (p = 0,0373), *ufd1l* (p = 0,0208; p<0,0001) or *mapk8a/b* (p = 0,0022) confirming their effect on ENS development (Fig 4A–C). Disruption of *slc8a1a/b* induced major phenotypical abnormalities including severe pericardial edema (S3 Fig), which has been reported previously for the *tremblor (tre)* mutant zebrafish (*NCX1h*, former gene name for *slc8a1a*) [48]. Larvae were *ufd1l* was disrupted failed to develop a swim bladder (S4 Fig), which was also reported previously for this mutant [49].

## Copy number loss alone is likely not sufficient to result in HSCR

Considering that aganglionosis has a low prevalence in patients with for example 22q11 [39, 40, 50] and 17q23 [51,52] deletions, CNVs alone are probably not sufficient to cause HSCR. HSCR penetrance is influenced by predisposing risk haplotypes in *RET*, semaphorin and *NRG1* [10–15]. Five non-coding risk haplotypes (Risk Score non-coding, RSnc; S6 Table) were tested in our three patient groups and compared to an *in-house* population dataset (group 5) (n = 727, RSnc = 2.54). As expected, HSCR groups 1, 2 and 3 all had a higher RSnc compared

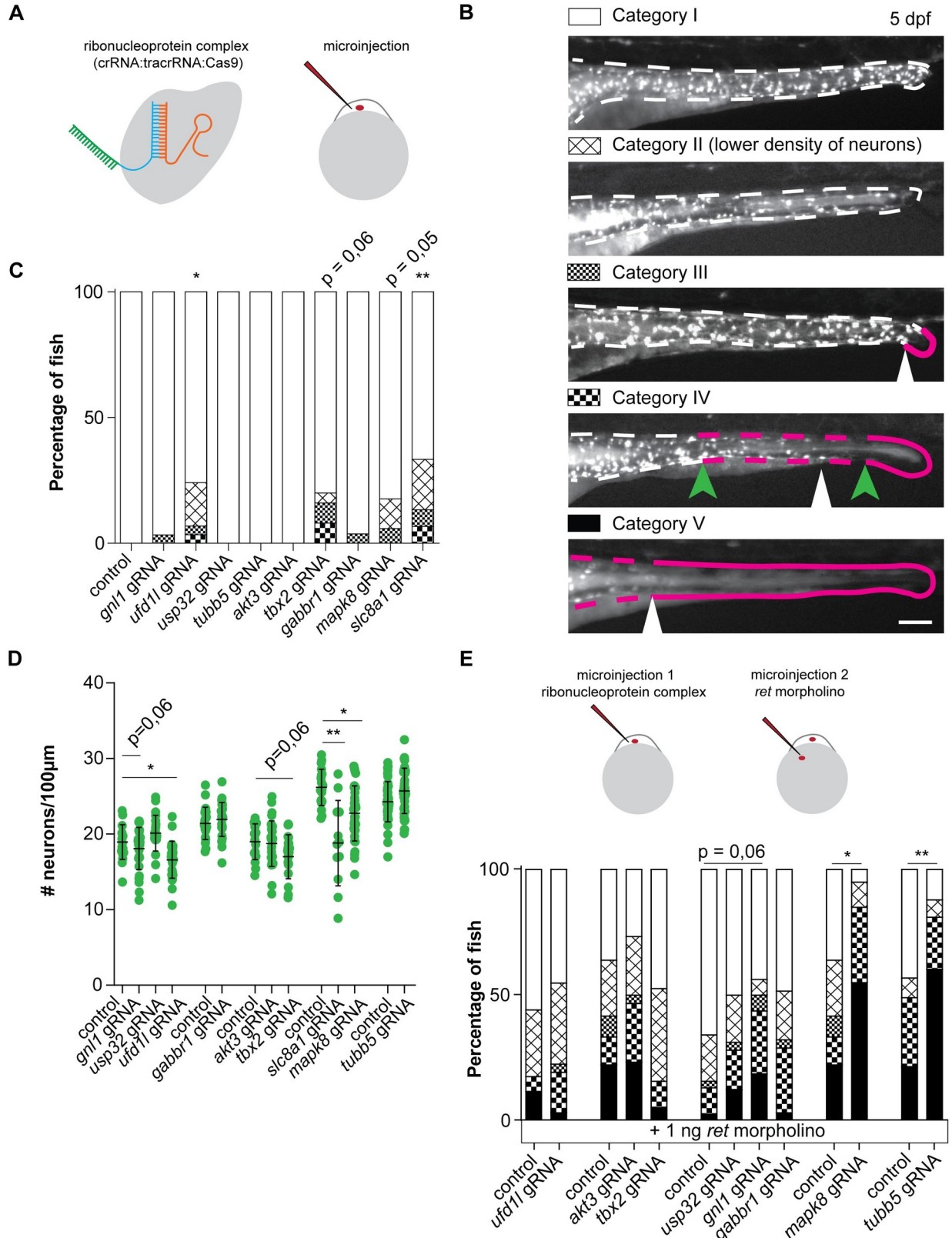

**Fig 3. Disruption of candidate genes in zebrafish caused defects in ENS development.** (A) Visual representation of the CRISPR/Cas9 complex injections using the Alt-R CRISPR-Cas9 System from Integrated DNA Technologies (IDT) [34]. (B) Legend showing the various phenotypes

observed in zebrafish. Larvae are characterized as category I (e.g. ENS not affected) when there is a full colonization of the intestine by phox2bb-GFP+ cells which are dispersed at a density comparable to that observed in untreated animals. Larvae are characterized as category II when there is an overall reduction observed in the presence of phox2bb-GFP+ cells along the total length of the intestine. Larvae are characterized as category III when an absence of phox2bb-GFP+ cells is observed in the most distal part of the gut, depicted by the white arrowhead and magenta line. Category IV classification is used when the phox2bb-GFP+ have migrated into the intestine but only reach in the mid-gut as in the example depicted by the white arrowhead. The green arrowheads as well as the magenta line indicate the range in which fish are categorized as category IV. The most severe alteration in the ENS is categorized as V, where the mid-gut and distal-gut are deprived of neurons, as depicted by the magenta line and the ENS has migrated no further than the white arrowhead. (C) Disruption of four genes induced ENS phenotypes in the zebrafish larvae, of which two were statistically significant (accumulated data from multiple experiments). (D) In line with the observed ENS phenotypes, quantification of the number of neurons per 100μm showed significantly reduced numbers of neurons upon disruption of three genes (accumulated data from multiple experiments). (E) Schematic representation of sequential injection with CRISPR/Cas9 complexes and ATG-blocking ret morpholino. Graph showing accumulated data of the percentage of fish with ENS phenotypes upon injection of a morpholino targeting Ret translation at a concentration that induced ENS phenotypes in approximately 50% of the fish. Ret morpholino injections in combination with disruption of mapk8a/b and tubb5 shows epistasis. Disruption of gnl1 showed a trend towards higher penetrance of ENS phenotypes. Statistical tests used: students t-test (D), one-way ANOVA followed by Dunnett's multiple comparisons test (D), and "N-1" Chi-squared test (C and E); * p < 0.05 ** p < 0.01 *** p < 0.001; Exact p-values, number of fish per group and statistics can be found in S11 Table.

to group 5 (p = 4.59825E-14, p = 0.000960603, p = 1.21875E-23, respectively) (S2 and S6 Tables), indicating a contribution of the risk haplotypes in all HSCR subgroups. Associated common risk haplotypes have epistatic interactions, not only with each other, but also with known HSCR genes -such as *RET* and *NRG1* [12,14,53]- and modify HSCR penetrance in monogenetic syndromes and Down syndrome [54–58]. The effect of epistatic interactions is smaller in high penetrant HSCR monogenetic disorders and larger in disorders in which HSCR penetrance is lower [54,55]. In line with this, contribution of these risk haplotypes was the lowest in group 2 (RSnc = 3.70), followed by group 1 (RSnc = 4.71), and the highest in group 3 (RSnc = 5.66) (S2 and S6 Tables). Thus, risk haplotypes have indeed a higher impact in HSCR patients without known coding variants, compared to those that contain known coding variants (S2 Fig and S6 and S7 Tables).

To model the reduced *RET* expression observed in patients carrying risk haplotypes, we injected zebrafish with an antisense ATG blocking morpholino (MO) specific for the zebrafish *ret* gene resulting in reduced Ret protein translation. We chose to inject 1ng of morpholino, as our previously published concentration curve showed that approximately half of the fish would present with an ENS phenotype (category II to V) [38]. Since disruption of *slc8a1a*/b in zebrafish resulted in severe heart epicardial edema and other malformations in ±75% of the surviving larvae and the absence of a swim bladder in all injected larvae (S3 Fig), we excluded this gene for the epistasis screen. Disruption of the candidate genes *mapk8a/b* (p = 0,0107) or *tubb5* (p = 0,003) in combination with the *ret* morpholino resulted in a significant increase of the percentage larvae that showed an ENS phenotype (Fig 3E). Considering that disruption of *tubb5* alone (Fig 3C and D) did not induce an ENS phenotype, these results suggest that loss of *tubb5* shows epistasis with *ret*. Disruption of *gnl1* also seemed to have an influence on the penetrance of ENS phenotypes in zebrafish (n.s.), therefore we repeated the injections using a lower dose of *ret* ATG blocking MO. Using a lower dose of MO results in a less severe phenotype, providing us with a more sensitive assay to detect epistasis between Ret and *gnl1*. Using this assay disruption of *gnl1* indeed showed a significant increase in penetrance of an ENS phenotype (p = 0,0405)(Fig 5A). Considering that disruption of *gnl1* alone did not induce an ENS phenotype, these results suggest that only in a sensitized background loss of *gnl1* can increase penetrance of ENS phenotypes.

## Discussion

We describe that HSCR-AAM patients contained more often large CNVs. Concordant with a previous study, specifically rare large CN losses were enriched in HSCR-AAM patients [45].

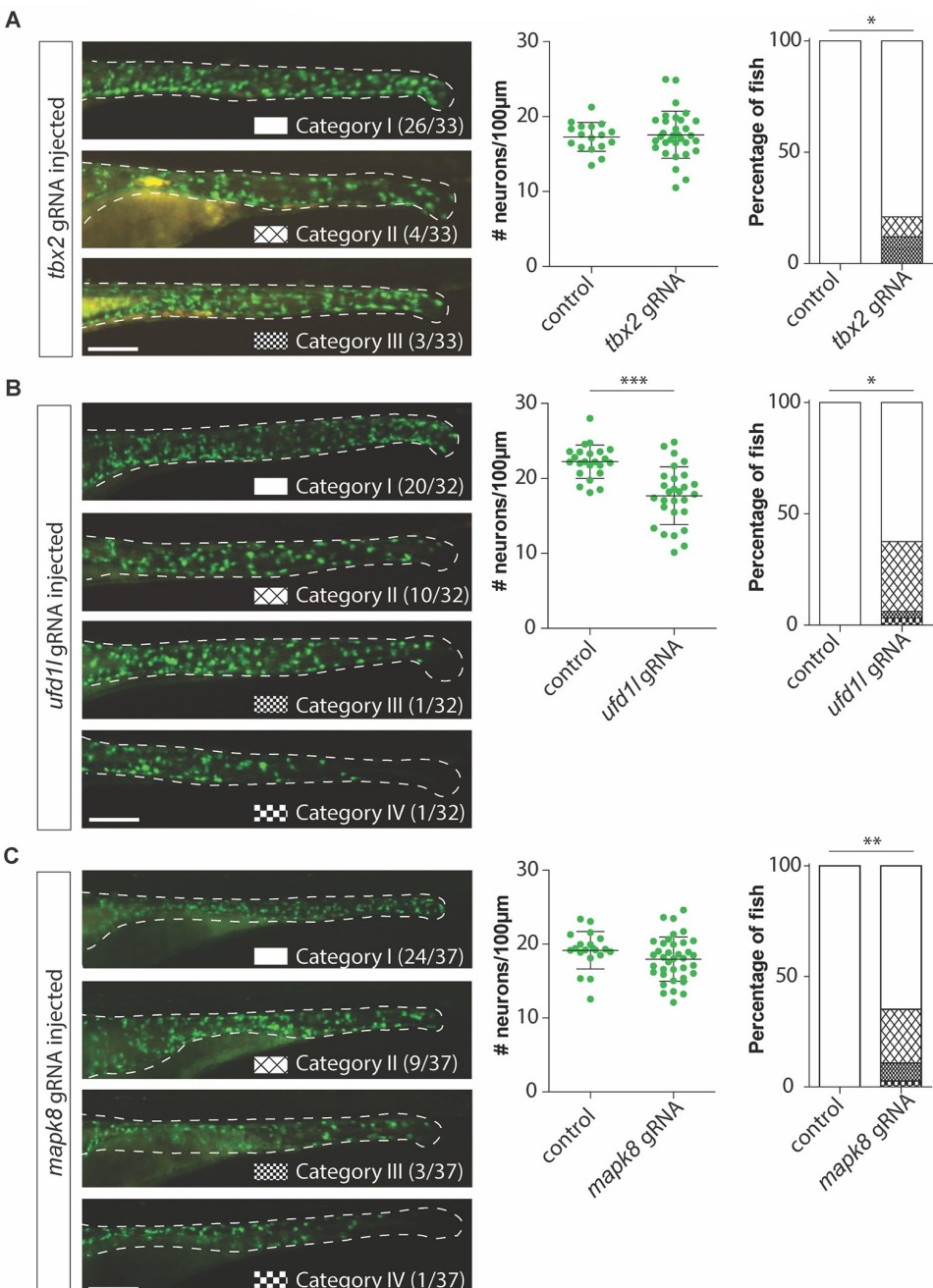

**Fig 4. Validation of tbx2, ufd1l and mapk8 gene disruption shows effects on the ENS in zebrafish.** (A) Figures and graphs showing that disruption of tbx2a/b in zebrafish does not result in overall significantly reduced numbers of phox2bb-GFP+ cells, however it does induce ENS phenotypes in category II and III in a significant number of zebrafish larvae. (B) Figures and graphs showing that disruption of ufd1l in zebrafish results in significantly reduced numbers of phox2bb-GFP+ cells and increase in the percentage of fish presenting with an ENS phenotype. (C) Figures and graphs showing that disruption of mapk8a/b in zebrafish does not result in overall significantly reduced numbers of phox2bb-GFP+ cells, however it does induce ENS phenotypes in a significant number of zebrafish larvae. Statistical tests used: students t-test and "N-1" Chi-squared test; * p < 0.05 ** p < 0.01 *** p < 0.001; Exact p-values and statistics can be found in S11 Table.

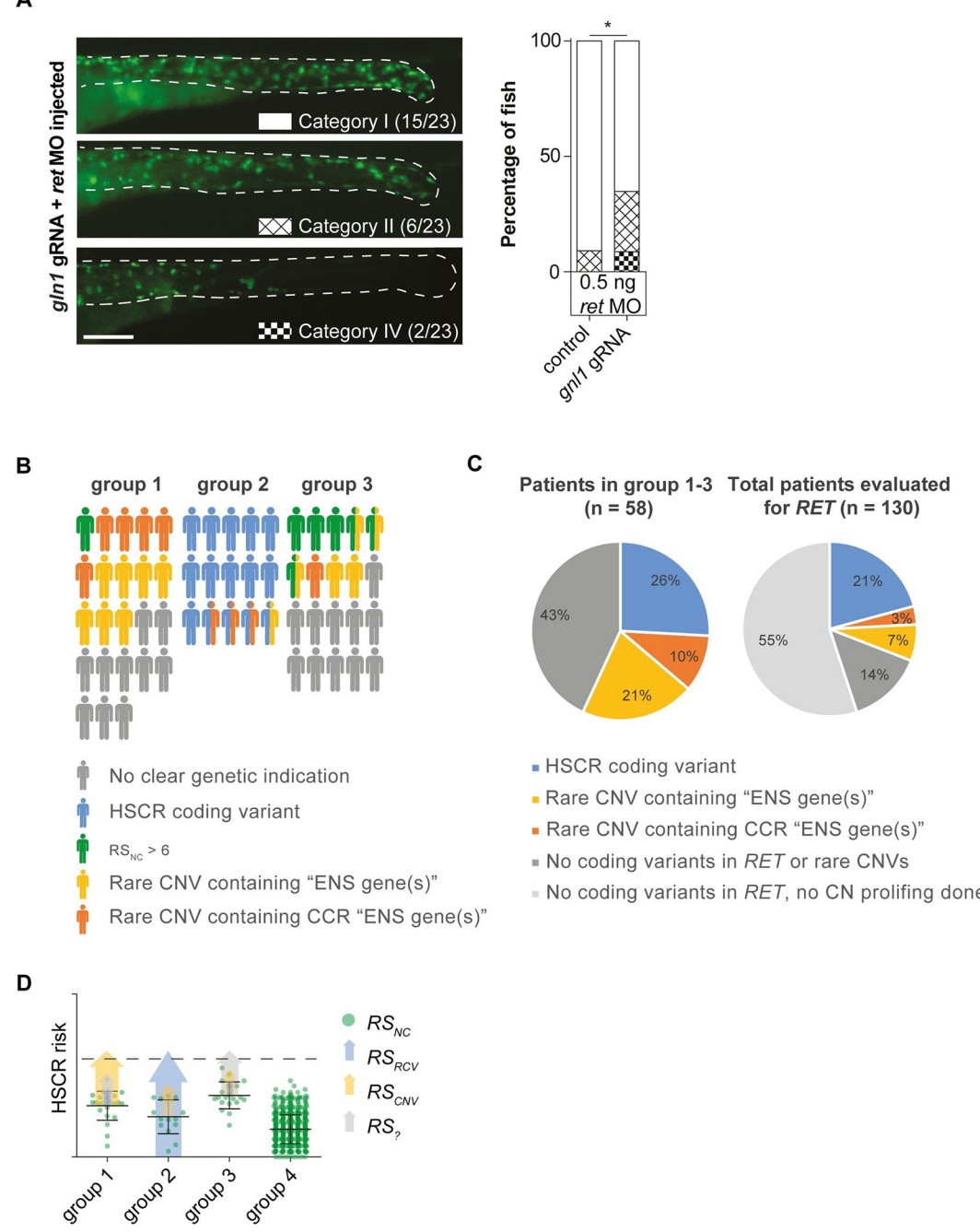

**Fig 5. Complex HSCR genetics: Epistasis between ret and gnl1 and genetic predispositions of HSCR patient groups.** (A) A second injection round targeting gnl1, using a lower dose of ret morpholino, confirmed gnl1 epistasis with Ret. Statistical tests used: "N-1" Chi-squared test; * p < 0.05 (B) Visual representation of the distribution of genetic predispositions over HSCR patient groups. In total 197 patients born between 1973 and 2018 were evaluated by a clinical geneticist in the department of Clinical Genetics, Erasmus Medical Center, Rotterdam. Of these, 114 did not have associated anomalies nor a known syndrome. 29 patients had a known HSCR related genetic syndrome, including Down syndrome (n = 18). 153 out of 197 patients were genetically evaluated for RET gene involvement and 21 had a pathogenic RET variant. (C) Pie charts showing the incidence of rare CNVs containing "ENS genes", CCR "ENS genes" and coding variants in HSCR patients. (D) Graphical representation of a hypothetical model explaining the relative contributions of the risk scores in our 3 patient groups. Error bars represent standard deviation.

We find that these rare CN losses contained more "ENS genes" that were a CCR. These characteristics make them suitable as candidate genes for HSCR. Therefore, we selected "ENS genes" that are CCR and were included in CN losses of HSCR patients. This resulted in nine candidate genes, which we disrupted in zebrafish larvae to screen for ENS phenotypes. CRISPR/CAS9 targeted knockdown of the candidate genes *SLC8A1*, *TUBB*, *GNL1*, *MAPK8*, *UFD1L* and *TBX2* found in five patients connects these genes to ENS development. In addition, for two genes we identified a second HSCR patient in which the gene was affected.

## Pinpointing the likely culprit in a CNV: Lessons from animal models

Mitogen-Activated Protein Kinase 8 (*MAPK8*) was impacted by the 10q11.22—q11.23 loss inherited from an unaffected mother. *MAPK8* is part of the Mitogen activated protein (MAP) kinase pathway and required for ENCCs to migrate properly in response to GDNF in mouse intestinal explants [59]. The RSnc in this patient is 4.57, indicating that—in addition to the predisposing common haplotypes—more (genetic) factors are needed to result in HSCR etiology of this patient. Zebrafish morphants for *mapk8b* (*jnk1*) have a ventralized phenotype, but their ENS was not investigated in this previous report [60]. Based on our zebrafish data presented here, an increased presence of larvae with an ENS phenotype was observed upon disruption of *mapk8a/b*, alone, and in epistasis with Ret. This suggests that loss of MAPK8 influences ENS development, and that this loss alone is sufficient to induce alterations in ENS development in zebrafish. However, more larvae are affected upon *ret* knockdown supporting the role of a combined effect of the common haplotypes and the deletion of *MAPK8* seen in this patient.

Patient P_000561 has the typical 22q11 deletion seen in 22q11 deletion syndrome. This deletion has a high penetrance with a variable phenotype [39,40,50]. HSCR has been previously described in patients with this deletion [61], as well as disturbances in migration of neural crest cells [62]. One of the main candidate genes for HSCR on 22q11 is Ubiquitin Recognition Factor In ER Associated Degradation 1 (*UFD1L*). It encodes for a downstream target of HAND2 [63]. *Hand2*−/− mice have decreased numbers of enteric neurons, neuronal differentiation defects and a disorganized ENS [64,65]. Mice with a targeted deletion encompassing *Ufd1l* as well as other genes of the 22q11 deletion syndrome present in humans, have shown to model the 22q11 syndrome [66,67]. However, the ENS was not evaluated in these latter models. Our zebrafish work shows that loss of *ufd1l* reduces the number of enteric neurons in a subset of zebrafish larvae. As zebrafish are "outbred" animals, it is likely that their genetic background in combination with loss of *ufd1l*, defines whether an ENS defect develops. This is in line with findings in patients, as HSCR is a rare phenomenon in 22q11 deletion syndrome [68]. Hence, it is likely that other factors contribute to HSCR development. Indeed, this patient (P_000561) has a RSnc of 4.90 suggesting a contribution of the predisposing haplotypes. It is likely that the relatively high impact of the common risk haplotypes, in combination with the *de novo* CNV impacting *UFD1L*, results in a shift from normal to defective ENS development.

Patient P_000567 has a *de novo* deletion of 17q23.1–23.2, a RSnc of 5.63 and has facial dysmorphism, hearing loss, microcephaly, immunological hypersensitivity, and nervus flammeus. This phenotype overlaps with other patients with 17q22-q23 deletions [51,52]. One of the other patients is constipated, but HSCR disease has not been previously described [69,70]. Four "ENS genes" are included in the CNV of patient P_000567: *BCAS3*, *HEATR6*, *TBX2*, *USP32*, of which the latter two are a CCR. Transient disruption of *usp32* in zebrafish did not induce changes in their ENS and yielded no indication that *USP32* is involved in HSCR. T-Box 2 (*TBX2*) codes for a transcription factor already known to be involved in the regulation of neural crest derived melanocytes [71], a process that is also hampered in specific HSCR

patients carrying pathogenic variants in *EDN3*, *SOX10* and *EDNRB* [72,73]. *Tbx2* heterozygous mice develop normally, but homozygous mice have severe cardiovascular defects [74]. Zebrafish *tbx2a/b* morphants also present with a cardiac phenotype as they show an expanded atrium and a smaller ventricle [75]. HSCR or gastrointestinal anomalies were not described, but likely also not evaluated, in these mice or fish. Disruption of *tbx2a/b* in our study resulted in increased incidence of ENS phenotypes, but the larvae did not present with overt cardiac phenotypes (S4 Fig), although we should note that we did not study this in detail. Taken together, it is likely that reduced *TBX2* expression has a role in HSCR development, yet only develops in patients with the 17q22-q23 deletion, if significant contributions of predisposing noncoding haplotypes are present, since this patient has a relatively high RSnc.

Patient P_000479 lost one copy of *SLC8A1* and has no associated anomalies. Additionally, a known pathogenic *RET* mutation: *RE*T: NM_020630: c.656-21C>T is present. As expected, the RSnc in this patient is low (RSnc = 2.98), which implies a strong influence of other genetic factor(s), in this case a pathogenic *RET* mutation. Whereas heterozygous *Slc8a1* knockout mice develop and mature normally, without evident phenotypical abnormalities [76]. *Slc8a1* homozygous knockout mice die *in utero* and have underdeveloped hearts [76]. In line with this, we show that zebrafish larvae where *slc8a1a/b* was disrupted, severe heart edema was present, as well as other morphological abnormalities (S3 Fig), as also reported in the *tremblor (tre)* mutant zebrafish [48]. In addition, a subgroup of zebrafish larvae showed aberrant ENS development, suggesting that this gene also affects ENS development. Since particularly, the *slc8a1a* isoform was efficiently targeted (84% efficiency, $R^2 = 9$), is likely that *slc8a1a* is the functional gene in zebrafish, in line with the *tre* mutant [48]. However, HSCR patients with a CN loss lose only one copy of *SLC8A1*, and the remaining gene expression could explain their milder phenotype. Together, this suggests that variation in the levels of SLC8A1 can have a major impact on various developmental processes and could very well play a role in the development of both HSCR and associated anomalies.

The 1q44 loss present in patient P_002431 overlaps with one of the losses described in DECIPHER patient (ID 249405). The only overlapping gene is AKT serine/threonine-protein kinase 3 (*AKT3*). Unfortunately, we were not able to confirm the effect of AKT3 deletion on disturbed ENS development.

Patient P_000512 with short segment HSCR, epilepsy and intellectual disability, has a large de novo deletion (6p22.1—p21.33) which affects the ENS genes *GABBR1*, *GNL1* and *TUBB*. The gamma-aminobutyric acid B receptor 1 (*GABBR1*) is the receptor for the inhibitory neurotransmitter GABA, and is expressed in the submucosal and myenteric plexus of the ileum and colon [77]. Receptor agonist studies indicate there is a GABA subtype and intestinal region-specific impact on gastrointestinal motility [78–82]. However, our zebrafish findings did not suggest a role for *gabbr1* in ENS development. Little is known about the Guanine nucleotide binding protein like 1 (*GNL1*) function. There are no mouse models described in the mouse genome informatics database [83](http://www.informatics.jax.org/). However, *in vitro* experiments indicate that GNL1 induces hyperphosphorylation of RB20, and promotes cell cycle progression and cell proliferation. GNL1 mRNA and protein expression in HIPED [84](https://www.genecards.org/) and GTEX [85](https://gtexportal.org/home/) respectively, highly correlates to that of kinesin family member 1-binding protein (KIF1BP). Recessive *KIF1BP* (formerly *KIAA1279*) pathogenic variants are involved in Goldberg-Shprintzen syndrome, often characterized by HSCR as well as intellectual disability [86,87]. Moreover, both genes (*GNL1* and *KIF1BP*) are "ENS genes" [41,42] and are expressed in human embryonic gut [88]. Disruption of *gnl1* in zebrafish showed epistasis with *ret*. The third candidate gene included in this patient its CNV is *TUBB*. Patients with mutations in *TUBB* have circumferential skin creases Kunze type (CSC-KT) which includes circumferential skin creases,

microcephaly, cleft palate, intellectual disability and other associated anomalies but also structural brain anomalies are described [89]. Symptoms in our patient are less severe as seen in patients with dominant negative missense mutations, most likely due to the fact that TUBB heterodimers can still form, albeit at reduced levels. Breuss and colleagues described heterozygous *Tubb5* mice as having decreased brain volumes due to increased apoptosis [90]. Another study in mice showed that perturbations to *TUBB5* have a deleterious effect on multiple aspects of neuronal differentiation (e.g. dendritic spine maturation, axon outgrowth) [91]. We did not find an effect on ENS development in zebrafish upon loss of *tubb5* alone, however, in combination with reduced Ret expression *tubb5* disruption significantly increased the incidence of an ENS phenotype. This is in line with the high RSnc of 5,95 observed in this patient. In addition, we found a mutation in *TUBB* in a patient from an independent HSCR cohort. Therefore, the absence of enteric neurons in the distal colon and other neurological symptoms seen in Patient P_000512, can be likely related to the deletion of *TUBB* and *GNL1* in combination with its high RSnc.

## Limitations, discrepancies and recommendations

Although we find differences in the size of the rare CNVs, we did not find differences in the overall presence of rare CNVs. However, a previous study described that HSCR patients with associated anomalies have more CNVs compared to isolated HSCR patients and unaffected controls [45]. An explanation for this discrepancy might be that we used a CNV frequency cut-off of 0.026%, to HSCR prevalence (0.02–0.03%). Therefore, these CNVs are too rare to find a significant association with our study cohort size.

We were able to pinpoint and associate several genes within a CN loss to the HSCR phenotype seen in these patients. However, this is most likely an underestimation of the true contribution of CNV to HSCR, since SNP-array has relative limited resolution to detect small CNVs and is not able to detect balanced structural variations as inversions and inversion-duplications. In addition, approximately 50% of the rare CNVs we found were CN gains. As only CN losses–not CN gains–proved to be significantly enriched in this HSCR-AAM cohort, we did not include CN gains in our screen. In future studies, we would recommend the use of whole genome sequencing (WGS) approaches, as these are able to confidently detect much more structural variant types and thus provide more patients with a definite genetic diagnosis. The benefits of WGS approaches would include the detection of inversion-deletions (and disrupted genes within a CN gain), more precise breakpoint mapping, the detection of balanced rearrangements (and disruption of TAD borders and/or regulatory elements by CN losses or gains), or a second mutation in a gene impacted by a CNV on the other allele (unmasking a recessive mutation or variant). Screening at improved resolution and ability to detect a wider range of genetic aberrations would likely increase the number of associated genes in the remaining patients without an explained genetic diagnosis.

## HSCR is a complex and heterogeneous genetic disorder

We determined segregation of the rare CNVs in nine patients, five of these were *de novo*. Although we did not determine the segregation of all rare CNVs, this already suggests the high frequency of *de novo* CNV in this cohort. In line with this, the DECIPHER (https://decipher. sanger.ac.uk/) contains 18 HSCR patients with associated anomalies with one or more CNVs. Of these, more than half of the CNVs are *de novo* (10 out of 18). Together, this shows that CNVs found in HSCR-AAM patients are often *de novo*. The results presented here highlight the importance of Copy Number profiling, leading us to propose a HSCR risk model that consists of risk scores for: rare coding variants (RSrcv), non-coding variants (RSnc), rare CNVs

(RScnv) and currently unknown risk factors (e.g. epigenetic modifications and perhaps environmental factors). Considering HSCR-AAM (group 1), 36% of patients contained a CNV harbouring "ENS genes" in CCRs (Fig 5B). Our risk model (Fig 5D) indicates the major contributing factors in each group: CNVs in group 1, coding variants in group 2 and noncoding variants in group 3 (Fig 5B and D). The predisposing *RET* haplotypes reduce RET expression [92] and in combination with the other risk haplotypes increase HSCR risk substantially (RSnc). Translating this population risk to a threshold for HSCR development in individual patients is challenging as it is not precisely known how many of (and if) these known predisposing haplotypes are sufficient to result in HSCR. If we set a threshold at a relatively high level (RSnc > 6), this would explain HSCR in 30% of patients in group 3 (Fig 5B and S7 Table). Most HSCR-AAM patients have a lower RSnc, which makes sense, as this group contains more high impact CNVs and whilst they lack *RET* coding variants, they potentially have variants in other genes (Fig 5B and S7 Table). Translating our findings back to our full cohort including all patients seen by a clinical geneticist and screened for *RET* mutations, HSCR coding variants explain 21% of HSCR cases (Fig 5C). If we consider CNVs containing "ENS genes" in a CCR, we could explain an additional 10% (Fig 5C). Importantly, this is an underestimation, since for 55% of patients without a *RET* coding variant no copy number profiling is performed (Fig 5C) and we focussed exclusively on rare CNVs. Since more common CNVs can also modify HSCR penetrance [45,93], our results emphasize the need for a more widespread genomic analyses in all subgroups. In addition, RSnc could be one of the determinants for a more elaborate genetic evaluation. A high RSnc in a HSCR patient without associated anomalies is less likely to have a deleterious RET coding variant, patients with a low RSnc are more likely to benefit from exome sequencing. Similarly, patients with large *de novo* losses often have a more complex phenotype and more intensive clinical investigations might be indicated.

## Conclusions

To summarize, HSCR genetics is complex with contributions of predisposing haplotypes in all HSCR subgroups. Rare large CNVs—often *de novo* (S1 Fig)—contribute substantially to the disease in HSCR-AAM patients. These CNVs are enriched for CN losses and for "ENS genes" that are also a CCR. Disruption of these genes in zebrafish confirmed that reduced expression of some of these genes increased the occurrence of ENS alterations, alone, or in epistasis with Ret. These genes have functional overlap with known HSCR disease genes: e.g. *UFD1L* is involved in signalling receptor binding and *MAPK8* in axon guidance. Our "ENS gene" based approach led to the identification of new HSCR candidate genes (Figs 1B and 3 and Table 3): *UFD1L, TBX2, SLC8A1, GNL1, TUBB* and *MAPK8*.

## Materials and methods

### Ethics statement

This project was approved by the Medical ethics committee of the Erasmus Medical Centre (Hirschsprung disease: no 2012–582, addendum No. 1 and no.193.948/2000/159, addendum No. 1 and 2, MEC-20122387). Written (parental) formal consent was obtained.

All authors had access to the study data and reviewed and approved the final manuscript.

### Determination of copy number variation

CNV profiles were determined with either the HumanCytoSNP-12 v2.1 or the Infinium Global Screening Array-24 v1.0 (Illumina Inc., San Diego, CA, USA), using methods, thresholds and

analysis settings previously described [94]. CNV profiles were inspected visually in Biodiscovery Nexus CN8.0 (Biodiscovery Inc., Hawthorne, CA, USA). CNVs with an overlap of at least 75% with similar state CN changes, were either classified as rare, when absent from large control cohorts (n = 19,584), or as a known modifier [39,40]. CNV number, size, type and gene content of rare CNVs were determined in HSCR patients (n = 58) and unaffected controls (n = 326) and compared between the control groups and the described HSCR subgroups (Fig 1A). All rare CNVs were uploaded to the ClinVar database (https://www.ncbi.nlm.nih.gov/clinvar/) and are depicted in S1 Table.

## Exclusion of the involvement of known disease genes

The presence of *RET* coding mutations and those in intron-exon boundaries, in all patients in this study are determined. Furthermore, if a specific monogenetic syndrome (S3 Table) was suspected, based on the phenotypic spectrum observed, the suspected gene(s) were evaluated using a targeted NGS panel or whole exome sequencing. In four HSCR patients with associated anomalies (Group 1) and nine HSCR patients without associated anomalies (Group 3), the involvement of other known disease genes was excluded [8,46,95,96] using whole exome sequencing (WES) with previously described pipelines [97,98] and variant prioritization methods [99].

## Evaluation of candidate gene expression

We prioritized candidate genes based on gene characteristics: genes that are intolerant to variation and/or dosage sensitive (CCR; see Variant prioritization in the next paragraph), and higher expressed in the developing mouse ENS compared to whole intestine, between embryonic day E11.5 and E15.5, referred to as "ENS gene" [41,42]. Genes affected by a rare CNV were extracted from the CNV regions and converted to mouse orthologues to make a gene panel for downstream analysis (S12 Table). Gene expression data used to determine the "ENS genes" was downloaded from the gene expression omnibus (GSE34208 and GSE100130). We calculated probe set summaries from the raw Affymetrix data using BRB-ArrayTools version 4.5.0—Beta_2 (http://linus.nci.nih.gov/BRB-ArrayTools.html) and used the "Just GCRMA" algorithm to adjusts for background intensities. We normalized each array using quantile normalization that includes variance stabilization and log2 transformation. Replicate spots within an array were averaged. Genes showing minimal variation across the set of arrays were excluded from the analysis (if less than 20% of expression data had at least a 1.5 fold change in either direction from a gene its median value, or at least 50% of arrays had missing data for that gene). The minimum fold change for the class comparisons was set at 1.5 and differential expression was determined using a two-sample T-test with random variance model. The permutation p-values for significant genes were computed based on 1,000 random permutations and the nominal significance level of each univariate test was set at 0.05. Subsequently, we defined "ENS genes" as genes higher expressed in isolated "ENS cells" (with and without the addition of GDNF) compared to the expression in the other intestinal cells or total intestines. These genes were considered as "ENS genes" and included in downstream analysis. As only micro-array-based expression data of the mouse ENS was available, we cannot reliably distinguish between not expressed and not differentially expressed genes between the ENS and the intestine. Unfortunately, there was no data available from human intestines during ENCC migration. However, RNA sequencing data from human colon was available for embryonic week 12, 14 and 16 (GSE111307) [88], we annotated the gene expression during these time points for reference to S12 Table. We used the settings and methods described previously, as well as the previously generated and publicly available datafile

"GSE111307_ENS_development_lenient_mapping.xlsx" at https://www.ncbi.nlm.nih.gov/geo/query/acc.cgi?acc=GSE111307.

## Variant prioritization in CCR

When determining if a gene affected by a rare putative deleterious CNV was a Constrained Coding Region (CCR) [43,44], we allowed some tolerance to account for reduced penetrance (filter settings and table see S12 Table). Variants from NGS data previously generated were prioritized according to settings described in S5 Table. These data included (1) a WES cohort of sporadic HSCR patients (n = 76, 149 controls) and (2) a Whole Genome Sequencing (WGS) cohort of 443 short segment HSCR patients and 493 unaffected controls [46]. Using RVTESTS [100], a variant burden test was done comparing the variant burden in 443 short segment HSCR patients and 493 controls (S5 Table). All rare putative deleterious loss of function variants unique to the HSCR cohort in CCR were described in Table 3.

## Genotyping of HSCR associated SNPs

Sanger sequencing was used to genotype all patients for SNPs known to be associated to HSCR [10–12,93]. Primer sequences can be found in S8 Table. We used (proxy) SNPs present on the GSAMD-v1 chip to determine the Rotterdam population background for these risk haplotypes (n = 388 females, 339 males), as well as the three HSCR subgroups from the SNP-array platform (S6 Table and S2 Fig). The relative weighted risk score of published common and relatively rare risk alleles near *RET* (rs2506030, rs7069590, rs2435357), *NRG1* (rs7005606) and *SEMA3C/D* (rs11766001, rs80227144), was calculated using the formula below [10–12,53].

$$((LnOR\ risk\ allele\ 1) \ast allele\ count) + ((LnOR\ risk\ allele\ 2) \ast allele\ count) + etc.$$

We also determined the risk scores of the CNV control samples (n = 326). However, we could only do this for the *RET* and *NRG1*, as no suitable proxy SNPs were present on the HCS 850k platform for the SEMA3C/D haplotypes (S2 Fig and S6 Table). Therefore, we used (proxy) SNPs present on the GSAMD-v1 chip to determine the Rotterdam population background for all risk haplotypes (group 5, n = 727, S2 Fig).

## Experimental animals

Zebrafish were kept on a 14h/10h light–dark cycle at 28˚C, during development and adulthood. Tg(*phox2bb*:GFP) animals were used for all experiments [47]. Larvae were kept in HEPES-buffered E3 medium and 0.003% 1-phenyl 2-thiourea (PTU) was added 24 hours post fertilization (hpf), to prevent pigmentation. All animal experiments were approved by the Animal Experimental Committee of the Erasmus MC Rotterdam.

## CRISPR/Cas9 gene disruption

Gene targeting using CRISPR/Cas9 was performed as described previously [35]. The Alt-R CRISPR-Cas9 System from Integrated DNA Technologies (IDT) was used (Fig 3A) [34]. crRNAs were designed using the IDT tool and selected on high target efficiency and low off target effect (S9 Table). crRNA(s) and tracrRNA were diluted to final concentration of 100uM in the provided duplex buffer. crRNA/tracrRNA complexes were generated by adding them in a 1:1 proportion, heating to 95 degrees Celsius for 5 minutes and let it cool down to room temperature. 1μl of crRNA/tracrRNA complex solution was used in an injection mixture with a total volume of 6 μl (in 300mM KCl) containing 4ng cas9 protein (Generated from addgene plasmid #62731) and 0.4 μl of 0.5% Phenol Red (Sigma-Aldrich). 1nl was injected into the

one-cell stage. gRNA sequences are listed in S9 Table, primers used for Sanger sequencing are listed in supplementary S10 Table. Efficiency of indel generation was determined as described previously (S11 Table) [35].

### Morpholino injections

Fertilized zebrafish eggs were injected with the CRISPR/Cas9 complex. Subsequently, 0,5 or 1 ng of translation blocking morpholino against *ret* (5′- ACACGATTCCCCGCGTACTTCCC AT -3′), and 1:10 0,5% Phenol Red (Sigma-Aldrich), was injected between the 1- and 4-cell stage (see Fig 3E) [101]. To minimize variability, the same needle was used to inject the controls, that were not injected with CRISPR/Cas9 complex and droplet size (injection volume) was checked regularly during injections.

### Imaging

Images of 5 dpf tg(*phox2bb*:GFP) larvae were taken using a Leica M165 fluorescent dissection microscope with the Leica LASX software. Larvae were anesthetized with 0.016% MS-222 in HEPES buffered E3 medium and positioned on their lateral side on a 1.5–2% agarose coated petridish, to enable visualization of the enteric neurons.

### Quantification of *phox2bb*:GFP+ enteric neurons

The number of *phox2bb*:GFP+ enteric neurons was determined using FIJI [102](ImageJ2, http://developer.imagej.net/about) by manual polygon selection of the intestine, duplication of the selected region, selection of the GFP channel and the find maxima function. The noise tolerance was adjusted manually to detect all neurons present but exclude noise. If needed, the cell numbers were manually corrected for cells that were missed by the tool. The length of the gut region included in the neuron count was measured using the straight-line tool. The number of neurons was normalized to the gut length and expressed in number of neurons per 100 μm.

### Statistical analysis

The number and size of rare CNVs, the number of rare losses and gains, the number of genes intolerant to variation (single nucleotide variants (SNVs) and CNVs), the number of "ENS genes" per rare CNV, and the relative weighted risk score, were determined and compared for the different groups with a single ANOVA test. If group differences existed (p < 0.05), we determined which subgroups were significantly different, using a two-tailed T-test. For statistical analysis of the zebrafish experiments the online tool MedCalc (MedCalc software ltd., Ostend, Belgium), was used together with the "N-1" Chi-squared test for the categorization. For the number of neurons we used a student's *t*-test (two groups per experiment), or one-way ANOVA followed by Dunnett's multiple comparisons test if the comparison included more than two groups per experiment, number of larvae used and p-values can be found in S11 Table. * p < 0.05 ** p < 0.01 *** p < 0.001.

### Supporting information

**S1 Fig. The Z-score distributions of total CNV size, CN loss and CN gain of controls and HSCR patients.** *Depicted are the Z-score distributions of all rare CNVs (black lines), rare CN losses (red lines) and CN gains (blue lines). Controls in continuous lines, all HSCR patients in dotted lines. Limited negative z-scores due to the size limit cut-off of 20kb. The z-score distributions of CNV sizes were comparable between groups strongly suggesting that the outliers are*

*responsible for the statistical significant groupwise-differences.*
(DOCX)

**S2 Fig. Risk score comparison between Sanger sequenced and SNP-array genotyped samples.** *Depicted are the calculated noncoding risk scores (RSnc) of patients using the genotypes derived from the 850K array platform. However, we could only do this for the RET and NRG1, as no suitable proxy SNPs were present on the HCS 850k platform for the SEMA3C/D haplotypes (S2 Fig). Therefore, we used (proxy) SNPs present on the GSAMD-v1 chip to determine the Rotterdam population background for all risk haplotypes (group 5, n = 727, Panel B in S2 Fig).*
(DOCX)

**S3 Fig. Brightfield images of zebrafish larvae injected with *slc8a1* gRNAs (*slc8a1a/slc8a1b*).** *Brightfield images of zebrafish larvae injected with gRNAs targeting slc8a1a/b showing severe phenotypes including heart edema, absence of swim bladder and small eyes. Scale bar = 500μm.*
(DOCX)

**S4 Fig. Brightfield images of zebrafish larvae injected with gRNAs.** *Brightfield images of zebrafish larvae injected with gRNAs targeting gnl1, tubb5, ufd1l, usp32, akt3a/b, tbx2a/b, gabbr1a/b and mapk8a/b showing mild phenotypes mainly consisting of the absence of a swim bladder and a down bend tail in case of ufd1l disruption. Scale bar = 500μm.*
(DOCX)

**S1 Table. Rare CNVs detected in this cohort.**
(DOCX)

**S2 Table. Statistical comparisons.** *The number and size of rare CNVs, the number of rare losses and gains, the number of genes considered a CCR, the number of ENS genes per rare CNV, and the relative weighted risk score, were determined and compared for the different groups with a single ANOVA test. If group differences existed (p<0.05), we determined which subgroups were significantly different, using a two-tailed T-test. Abbreviations: CCR; Constrained Coding Region; SNV; single nucleotide variant, CNV; Copy Number Variation, ENS; Enteric Nervous System, RSnc; Risk Score non coding, ND; not determined. Higher values in green, lower values in red. Two-tailed p-values.*
(DOCX)

**S3 Table. HSCR disease gene characteristics.** *Most known HSCR genes are intolerant to genetic variation and are rarely impacted by CNVs in unaffected individuals* [39, 40]. *These genes have been described to be impacted by CNVs in HSCR patients* [7, 27], *although this does not seem to be a frequent phenomenon, as in our cohort we did not detect any CNV impacting a known HSCR gene. Abbreviations: mis_z: Missense variation Z-score, syn_z: Synonymous variation z-sore, PLI: probability of being loss-of-function intolerant, PRec: probability of being tolerant to heterozygous loss of function variation but intolerant to homozygous loss of function variation, PNull: probability of being tolerant to loss of function variation (heterozygous or homozygous), del: deletion, dup: duplication, EW: embryonic week, DDD: deciphering developmental disorders project. Data derived from (*https://gnomad.broadinstitute.org/) *and the DDD control track at*: https://genome-euro.ucsc.edu/.
(DOCX)

**S4 Table. Rare CNV with "ENS genes".**
(DOCX)

**S5 Table. Variant prioritization and WGS variant burden test results in genes impacted by rare CNV.** *Abbreviations*: NVAR: *number of variants*, NCASEHET: *number of heterozygous*

variants in cases, NCTRLHET: number of heterozygous variants in controls.
(DOCX)

**S6 Table. Risk haplotype markers and odds ratio's used to calculate noncoding risk scores (RSnc).** *Highlighted SNPS in* **grey/ bold** *are the risk alleles and Odds ratio's used in the polygenic risk score calculation. #We did not account for the increased risk of having the two main risk haplotype combinations* [1]*. @ this risk haplotype was not present in this patient cohort. Sanger sequencing was used to genotype all patients for SNPs known to be associated to HSCR* [9, 10, 12, 13]*. Patient and controls genotypes were determined using SNPs from the GSAMD-v1 platform. D' and R2 derived from* (https://ldlink.nci.nih.gov/) *in European population. Primer sequences can be found in* S8 Table.
(DOCX)

**S7 Table. Description of individual patient genetic risk profiles.** *Depicted are the individual patient's characteristics, number of predisposing haplotypes (as represented by the risk SNPs) and the noncoding risk score (RSnc). Additionally, the RET risk haplotypes as described in* S6 Table *are depicted. In blue the number of "ENS genes" in a CN gain, in red the number of "ENS genes" in a CN loss. Depicted in blue: (1) functional evidence from zebrafish studies and (2) the number of additional patients containing putative deleterious variants in genes within the rare CNVs.* $^{\$}$*No phasing was performed to discern the most likely haplotype.*
(DOCX)

**S8 Table. Primer sequences common predisposing SNPs.**
(DOCX)

**S9 Table. Details of the gRNA sequences used.**
(DOCX)

**S10 Table. Primers used for Sanger sequencing.**
(DOCX)

**S11 Table. Statistics, group sizes and gRNA efficiencies of zebrafish experiments.**
(DOCX)

**S12 Table. Gene content of rare CNV.** *This list contains all genes in rare CNVs found in the four groups. We determined if a gene was a conserved coding region using the following criteria: sequence variants: PLI$\geq$0.85, z-score for missense variants and/or synonymous variants $\geq$3. Deletions or duplications: del score $\geq$ 1, dup score $\geq$ 1 and cnv score $\geq$ 1. Additionally, genes impacted more than once by a deletion, or a duplication were not considered a CCR. If these criteria are met, this gene is marked YES in the CCR column. We determined whether the gene is a "ENS gene" based on if the gene expression is higher in isolated "ENS cells" (with and without the addition of GDNF) compared to the expression in other intestinal cells or total intestines in mice* [41,42]*. These genes are marked YES in the "Mouse ENS E11-E15.5 gene" column. Filtering on CCR, "ENS gene" and CN loss results in our candidate gene list. Note that in addition to the nine genes we disrupted in zebrafish DPCR1 and ZSCAN31 are included. These genes were excluded as they do not contain a zebrafish orthologue as depicted in the second sheet "zebrafish orthologues". Sheet 3 "count tables" contains the count values used for* Fig 2.
(XLSX)

## Acknowledgments

We thank Mike Broeders for supplying us with the Cas9 protein.

## Author Contributions

**Conceptualization:** Alice S. Brooks, Robert M. W. Hofstra, Erwin Brosens.

**Data curation:** Erwin Brosens.

**Formal analysis:** Laura E. Kuil, Katherine C. MacKenzie, Clara S. Tang, Anwarul Karim, Bianca M. de Graaf, Erwin Brosens.

**Funding acquisition:** Clara S. Tang, Paul K. H. Tam, Robert M. W. Hofstra.

**Investigation:** Laura E. Kuil, Jonathan D. Windster, Thuy Linh Le, Bianca M. de Graaf, Robert van der Helm, Yolande van Bever, Cornelius E. J. Sloots, Conny Meeussen, René M. H. Wijnen.

**Supervision:** Maria M. Alves, Robert M. W. Hofstra, Erwin Brosens.

**Visualization:** Laura E. Kuil, Katherine C. MacKenzie, Erwin Brosens.

**Writing – original draft:** Laura E. Kuil, Katherine C. MacKenzie, Erwin Brosens.

**Writing – review & editing:** Laura E. Kuil, Katherine C. MacKenzie, Dick Tibboel, Annelies de Klein, Jeanne Amiel, Stanislas Lyonnet, Maria-Mercè Garcia-Barcelo, Paul K. H. Tam, Maria M. Alves, Robert M. W. Hofstra, Erwin Brosens.

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
