## [Decision Letter · Decision Letter 0]

5 Dec 2020

Dear Dr Brosens,

Thank you very much for submitting your Research Article entitled 'Size matters: large copy number losses reveal novel Hirschsprung disease genes' to PLOS Genetics.

The manuscript was fully evaluated at the editorial level and by two independent peer reviewers. As you will see, the reviewers appreciated the attention to an important problem, but raised substantial and serious concerns about presentation, organization, and annotation. Although there are no fundamental flaws with the approach or design, the editors agree with the reviewers that comprehensive reorganization and rewriting would be necessary for a revised manuscript to move forward. 

 Based on the reviews, we will not be able to accept this version of the manuscript, but we would be willing to review a much-revised version. We caution that the extent of effort necessary to satisfy the reviewers' concerns is considerable, and that for a revised manuscript to succeed, the reviewers' concerns will need to be satisfied fully.

If you decide to revise the manuscript for further consideration at PLOS Genetics, please aim to resubmit within the next 60 days, unless it will take extra time to address the concerns of the reviewers, in which case we would appreciate an expected resubmission date by email to plosgenetics@plos.org.

[LINK]

We are sorry that we cannot be more positive about your manuscript at this stage. Please do not hesitate to contact us if you have any concerns or questions.

Yours sincerely,

Gregory S. Barsh

Editor-in-Chief

PLOS Genetics

Gregory Copenhaver

Editor-in-Chief

PLOS Genetics

Reviewer's Responses to Questions

**Comments to the Authors:**

Reviewer #1: Because of the complexity of Hirschsprung Disease (HSCR) genetics, identifying the genetic alterations responsible for the condition is challenging. To explore whether rare CNVs contribute to aganglionosis in HSCR-AAM patents (HSCR with additional anatomical malformations) and to uncover genes underlying HSCR-AAM, the authors have combined analysis of patient CNVs, evaluation of CN loss regions for genes expressed in the embryonic mammalian enteric nervous system (ENS) to identify candidate genes influencing disease, and then testing the function of these candidate genes in ENS development in zebrafish. The identification of candidate HSCR disease associated genes is of interest to the fields of human genetics and developmental biology, however the current manuscript suffers from some significant organizational/readability issues as well as issues with the description, analysis and interpretation of the gene functional studies.

Major comments

1. The information needed to understand the experimental design and results is not where it needs to be and instead is buried within the supplemental material or in the methods, including some of the main findings. The main figures and supplementary figures lack appropriate legends. And the supplementary material is not well organized and much of it is not explicitly referred to in the paper. Some examples:

a) The rare CNV analysis is not properly introduced. Key to this analysis is the classification of subgroups of HSCR-AAM. Figure 1A/B/3, depict HSCR patients classification into 3 groups, however the results and figure legends never describe these groups. This information is found only in the “Patient Inclusion” section of the methods on page 8, and needing to search for this important information affects readability.

b) The evidence for the first finding, “The number of rare CNVs per individual did not differ between subgroups or controls (S4).”, appears to not be represented in the figure, but only in the S4 legend text. This is one of the key findings cited in the abstract, and should be included in the main text and/or be clearly represented by a figure data panel (i.e. S4n), instead of needing to hunt for the information. Likewise, data for another main finding cited on p4 “In Group 1, rare CNVs are enriched for ENS genes (p=4.565E-6, S7).” appears not to be within S7, but in S4, but again only in the figure legend. And it isn’t clear what data this point refers to. Again, these main findings should be clearly presented.

c) Further CNV analysis is found over the 1 table and 10 plots of S4. Each piece of data should have designators (i.e. S4n) and the finding from each plot indicated in the results section. Many of these results are currently only discussed within this supplementary figure legend. If these results are not presented in the paper, then they should not be included. The same is true for all data figures and supplemental data figures. Careful consideration must be made of what to include and to properly describe any included data.

d) Acronyms such as RS (and RSnc, RScnv and RSrcv) are not defined in the legends. Nor do they appear in the text on first appearance (i.e. not for Table 1, S4). RSnc eventually described in reference to Fig2C and S11 (see also point 1, above). The other acronyms are never described or referred to other than appearing in Fig1b. In Tables 1 and 2, the acronyms in the column headers are not defined. A few are also mentioned in S9, but still not defined.

e) The term “ENS genes” is used in the results on page 4 in reference to S7 (and in S4) but without explanation. After searching, a partial definition of this term is found within the legend of S4 and also obliquely described in the methods page 9. This is a key piece of information that should be clearly described in the relevant part of the results section.

2. Several issues relate to the ENS phenotypes described following CRISPR/Cas9 targeting of zebrafish genes:

a) Phenotypes are described variously as “hypoganglionosis”, “ultra-short segment HSCR”, “short segment HSCR” and “total colonic HSCR”. This is not a valid description of these zebrafish phenotypes. There is no justification for the use of these terms and they are highly misleading. The whole zebrafish intestine is not equivalent to the mammalian colon. Instead, phenotypes affecting the distribution of ENS neurons along the gut length should be presented in neutral terms/values that actually describe the phenotypes. While the term “hypoganglionosis” has been used previously to describe zebrafish ENS phenotypes with a reduction in ENS neuron number, this term is not accurate. The zebrafish ENS has no ganglia. A proper descriptive term should be used.

b) It is not stated how a designation of “hypoganglionosis” is given. Are neurons counted? What percentage reduction constitutes a phenotype?

c) Are phenotypes observed outside of the ENS? Are any growth abnormalities observed in any gene targeting experiment? This is an important point since general developmental delays would affect the number and distribution of neurons observed at 5dpf.

d) P-values are given for observed phenotypes in S16. It is unclear whether all types of phenotypes have been lumped together for these statistics.

e) Results and p-values are described for "tbx2" in Figure 2D and S16, but it is unclear if this refers to CRISPR/Cas9 targeting of tbx2a or tbx2b or a combination. The same question arises for gabbr1a/1b, slc8a1a/1b and mapk8a/8b.

f) Ns are low for all CRISPR/Cas9 experiments, ranging from 15 to 40 embryos per gene/gene pair targeted. In the case of slc8a1, a p-value of 0.0073 is given for an experiment involving 15 embryos. Power calculations should be shown to demonstrate the appropriate number of animals needed for statistically relevant results, and that the numbers used in these experiments are sufficient.

g) The description of the results in Figures 2E and 2F is very sparse and requires further elaboration and discussion.

h) The phenotypes observed in zebrafish should never be described as “HSCR” (i.e p7 of the Conclusions). See point 1a, above.

i) Candidate HSCR disease genes should be described as “candidates” rather than “HSCR disease genes” (see p. 8, for example)

j) A p-value is given for ret+tbx2, but no results are shown in Fig2e and no data is shown for slc8a1

3. Other issues:

a) Figure 1 is a graphical display of the experimental design. The images chosen are mostly far too small to see. Key information is missing (i.e. what are the subgroups, see point 1a, above), and some images are possibly not appropriate (‘ENS’ panel suggests sequencing gels were run? Wasn't this panel referring to microarray data?). This entire figure requires significant improvements to be useful.

b) Why isn’t ChAT chosen for analysis (Table 1)? From the logic presented, it’s not clear why this gene is not selected, for it features in S7 and is an ENS expressed gene.

c) The statistics designations (i.e. meaning of ** vs ***) are never explained in the methods. The actual statistics should be found in the figure or in the figure legend, in addition to the main text. This is especially important because some results are not described in the text (i.e. most of S4 plots where * designations are not given, though p-values are indicated in the legend. But it’s unclear which p-value is referring to which panel - see point 1c, above).

d) The term “overexpressed” is used many times, for example in “a constraint gene overexpressed in the mouse ENS” (p.9) , but these genes are NOT overexpressed in the ENS. This term suggests some alteration or manipulation. Instead, these genes are expressed in the ENS. They are also expressed in the ENS more abundantly than other reference tissues (i.e. other gut tissues). This term needs to be changed throughout the text/figures/legends/data tables, etc..

e) On p5 the statement: “However, there was no significant enrichment for nonsense, splice site or missense variants in genes impacted by rare gains nor losses (S9)" seems at odds with the data in S9, which shows p-values of <0.05 for 5 genes.

f) S15 appears to refer to figures incorrectly.

g) Attention to writing is needed throughout, and especially in the author summary.

Reviewer #2: This work presented in this manuscript involves screening of a population of HSCR patients with and without additional presenting phenotypic features for the presence of rare CNVs as compared to control individuals. CNVs were then filtered for predicted deleterious impact based on the ENS expression for genes located within the CNV and a gene’s intolerance to variation within coding regions. Candidate genes were subsequently functionally evaluated for a role in ENS development utilizing a zebrafish CRIPR/cas9 knockdown, ret morpholino sensitized model.

This article would immensely benefit form a comprehensive revision and organization of the material presented. Paper is not well organized and critical information is buried in the supplemental files without proper descriptions and annotation links from within the main paper. The paper would benefit from editing for verb tense and typographical errors. The manuscript as currently presented detracts from the data presentation.

From the start of the results section, the authors should clearly define the patient grouping characteristics upfront. This identification needs to be clearly stated in the main text and in the appropriate figure legends, not merely placed at the end, in the methods, patient inclusion section.

With respect to the assessment of gene function in zebrafish, full images representing gene knockdown embryos should be included in supplement as this data would be of interest to a broader audience. I would note that under the supplement section for patient P_000479, the authors do state “In line with this, we show that zebrafish larvae where slc8a1a and slc8a1b were disrupted, severe heart edema was present, as well as swollen heads”, however images of this data was not presented.

Authors should clearly note which of the two alleles for each gene are being displayed in the Figure 2f gRNAs plots. For each of these genes the authors may find helpful the ZFIN resource ( https://zfin.org/ ). This resource will allow for identification of published results of previously characterized Zebrafish gene expression and allele specific phenotypic results for comparison. These phenotypes should be included when discussing the gene expression or phenotypes alongside those of mouse models as the authors are utilizing zebrafish as their research model for phenotypic evaluation.

In supplemental section 8, authors utilize this section to not only present patient phenotypic information, but also expand on data presented and draw conclusions based on this data. Much of the information included here would benefit from a rewrite consolidation and reworking into the discussion of the main paper.

Authors should address in their discussion the large number of individuals remaining without an explained genetic diagnosis following this analysis including the extent to which complex rearrangements will remain undetected by SNP array-based analysis (eg. complex rearrangements without copy number alterations such as inversions and the need for better resolution of breakpoint junctions with respect to genes in the cases of duplications). The number of patients without definitive diagnosis is not unique to HSCR patients and limitations of the approach taken should at least be acknowledged. While authors have focused on rare CVNs with associated with region loss, roughly 50% of the rare CNV’s detected in table S3 represent duplications. Are these duplications included in the data for figure 3b and if so, do the authors have evidence of gene disruption at breakpoints or that upregulation of critical genes contained within the intervals contribute to HSCR?

Minor points.

Authors use the term “constrain” and “constraint” inappropriately throughout the text. These instances should either use the terms “genes under constraint” or “constrained genes”.

All figures should be numbered to reflect the order that they first appear in the text. For example, supplements S1 and S2 are not mentioned in the main text.

Figure 2. All asterisks in the figures should be appropriated defined in the respective figure legends.

Figure (2a). Label y- axis clearly, is it bp or Mb?

Figure (2d and e). The total number of animals used should be included in this figure/figure legend. At a minimum, authors should indicate within the figure 2 legend that the total number of animals evaluated is presented in S16. In addition, authors should clarify if the P-value presented in figure 2e is reflective of the asterisk to the right or does it indicate a P-value for the bars below.

Figure (2f). In the top panel, the fish are labeled “normal”, however this is then presented with a number 15/23. I assume the authors wish to convey that these are the “control” individuals with a normal distribution pattern. This should be better described in the figure legend.

The term CN should be defined in the text at first usage.

S4) A subset of graphs presented here appear to be the same as those already included in Figures 2a, 2b and 2c.

S6) Please define the headers for this table.

S16) Consistent capitalization/non-capitalization should be utilized (“p”-value and “chi” in S16 , and supplement index).

**Have all data underlying the figures and results presented in the manuscript been provided?**

Reviewer #1: Yes

Reviewer #2: Yes

PLOS authors have the option to publish the peer review history of their article (what does this mean?). If published, this will include your full peer review and any attached files.

Reviewer #1: No

Reviewer #2: No

---

## [Decision Letter · Decision Letter 1]

16 Mar 2021

Dear Dr Brosens,

Thank you very much for submitting your Research Article entitled 'Size matters: large copy number losses in Hirschsprung disease patients reveal genes involved in enteric nervous system development.' to PLOS Genetics.

The revised manuscript was seen by the two reviewers of the original submission. As you will see, both reviewers appreciate the changes that were made; however, there are some serious remaining concerns that will need to be addressed for the manuscript to move forward. Most of these concerns have to do with presentation and analysis, and are straightforward though not trivial to address.

We therefore ask you to modify the manuscript according to the review recommendations. Your revisions should address the specific points made by each reviewer. We hope to receive your revised manuscript within the next 60 days. If you anticipate any delay in its return, we would ask you to let us know the expected resubmission date by email to plosgenetics@plos.org.

[LINK]

Yours sincerely,

Gregory S. Barsh

Editor-in-Chief

PLOS Genetics

Gregory Copenhaver

Editor-in-Chief

PLOS Genetics

Reviewer's Responses to Questions

**Comments to the Authors:**

Reviewer #1: The revisions to the text and figure organization have improved this version of the manuscript. However, several points remain outstanding. These impact significantly on logic and readability and/or on data findings:

1) The terms “hypoganglionosis”, “ultra-short segment HSCR”, “short segment HSCR” and “total colonic HSCR” do not belong in the Results or the Figures. The fact that these authors have published their proposed classification strategy in a Review article (Kuil 2020) is immaterial to the point made here. This classification is not accurate or justified (the whole zebrafish intestine is not equivalent to the mammalian colon), is misleading and does not belong in a research paper Results section. Neutral, descriptive terms should be used in the Results to describe the findings. If the authors wish to raise in the Discussion the possibility that zebrafish phenotype severity may relate to HSCR phenotype classifications, and allude to their review article, this would be acceptable. The use of these terms is not only provocative and inaccurate, but also entirely unnecessary for this study, since the effect of gene disruptions are only described as having “significant effect” or “effect” or “significant increase in percentage of larvae that showed a phenotype”. The use of inappropriate terms offers no additional information and only clouds a simple analysis. Further to this general point, the terms “HSCR penetrance in zebrafish” and “HSCR penetrance” should not be used in the Results. The Discussion is the appropriate place to raise these speculative comparisons.

2) The term “overexpressed” is not accurate to describe genes more abundantly expressed in the ENS relative to a reference tissue, and should not be used. These genes are “expressed” in the ENS. The changes made to the manuscript do not address the issue raised.

3) In Figure 1, the schematic representation of the study set-up, the image used in Fig 1B to represent microarrays appears to be of a sequencing gel, though the legend states that this is intended to represent a heat map. This image is unrecognizable as a heat map. It is likely that the authors did not amend the figure to include the BRB-ArrayTools image, as was indicated in their rebuttal letter. Importantly, were heat maps used to identify differentially expressed genes? If so, this analysis should be shown. The methods suggest that straightforward differential expression analysis was used, presumably with some (unspecified) fold-change cut-offs, in which case heat maps should be shown to depict those genes that met this fold change cut-off, with an accompanying data table. Or perhaps scatterplots with fold change cut-offs shown and an accompanying data table. In the schematic, another more appropriate image should be found to represent microarray analysis, and best would be to select an image that reflects the analysis that was actually done. And any relevant analysis should be shown as a figure and/or data table. And the methods should be properly described in the Methods section. See also points 5 and 6 below.

4) The term “ENS gene” is still used before being defined (p. 8, lines: 170, 172, 177, 181), meaning that the logic and readability is still an issue in this version. Is this term even intended to be the same as “ENS gene” as defined on p9? Or is there some other criteria for the term used on p8? If different, this term should also be explained. If the terms are the same, then the definition must come earlier in the text. Moreover, the term as currently defined on p9 is not clear. Presumably an “ENS gene” is one which has higher expression in the developing mouse or human ENS compared with the whole intestine. As written, the definition refers to a sentence also describing CNVs, which is unnecessarily confusing: “Whereas a total of 230 genes affected by a rare CNV, had higher mRNA expression in the developing mouse ENS compared to the whole intestine (n=91 in group 1-3, n=139 in group 4) [43, 44]. We considered these genes to be important for ENS development and will refer to those genes as “ENS genes” throughout this manuscript.” This definition should be simply and clearly stated.

5) The section “Candidate genes with a loss are under constraint and expressed during ENS development” is very confusingly written. The first indication that expression of genes is being examined comes from: “Of these, 514 did not have a known mouse orthologue or probes in the microarray datasets used”. But nowhere is the use of microarray datasets explained in the Results. Certainly the schematic representation of the study set-up in Figure 1 does not adequately present or describe this part of the study (see also point 3, above) and nor does the cursory figure legend text: “The genes included in the CNVs were ranked based on their expression in the developing mouse ENS relative to that of the total intestine (for illustrative purposes a screenshot of a heatmap of differential gene expression derived from brb-array tools (https://brb.nci.nih.gov/BRB-ArrayTools/ )”. Information to understand this study belongs in Results section (see point 3, above). There is no reason not to describe this work simply and clearly.

6) There is no data table corresponding to the “Evaluation of candidate gene expression” to show the complete list of genes this analysis identified as expressed in the mouse ENS or the human embryonic colon. See point 3, above. This is required to show that the analysis was performed correctly and, importantly, to show qualitative expression information for the ENS expressed genes followed up in this study.

7) There is no experimental basis for speculating a role for TUBB in HSCR, since no effect was seen in Crispr/CAS9 experiments targeting tubb1. Therefore, this cannot be stated as is written (p16): “Although we did not find an (e)affect on ENS development in zebrafish, the absence of enteric neurons in the distal colon and other neurological symptoms seen in Patient P_000512, can be likely related to the deletion of TUBB and subsequently to an increased neuronal apoptosis or perturbed neuronal migration or differentiation in combination with the effects of GNL1.” Importantly, in fact, the zebrafish orthologue of TUBB is tubb5, although tubb1 (the TUBB1 orthologue) was tested in these experiments. So it appears that the right experiment has not been done in any case. If so, this would suggest that no conclusion can be made on the role of TUBB in HSCR and the Results and Discussion should be modified accordingly. Finally, the Discussion (in general) is very long, and especially so in this section discussing the lack of evidence for roles of GABBR1 and TUBB.

8) A table should be provided to show the orthologous Mouse and Zebrafish genes for the 1216 Human genes or transcripts present in CNVs of group 1-3 and group 4, to include the supporting source reference for these orthologue assignments (i.e. HGNC, MGI, ZFIN, Ensembl). This is important for point 7, above. Moreover, together with the table indicated in point 6, this information is necessary to provide support for statements such as “Of these, 514 did not have a known mouse orthologue or probes in the microarray datasets used” (p.9, line 201). Given the fact that 99% of human genes have mouse orthologues, it is unlikely that ~half of the 1216 human genes or transcripts lack mouse orthologues, and more likely that the many of the orthologous mouse genes were not present on the arrays. The table should provide information to allow these two possibilities to be clearly distinguished. This should also be discussed in the “Limitations and discrepancies” section.

9) The authors have addressed a comment from Reviewer 2 regarding “constraint”, but in doing so have now made further syntax/grammatical errors: i.e. “genes under constrained” throughout (i.e. Author Summary, Results, Discussion). In general the Author Summary continues to require attention to writing. Punctuation and other grammatical/syntax errors are an issue throughout the entire text and affects readability. Note also “Danio rerio”. The Discussion would benefit from shortening.

Reviewer #2: The authors resubmitted manuscript has undergone a significant rewrite of the manuscript including the approach they undertook, figures, and interpretation of results. This substantial rewrite now provides a clear presentation of the CNV’s identified, associated clinical presentations and analysis of genes located within these regions with respect to enteric nervous system expression and functional validation in zebrafish.

I would suggest the following minor comments be addressed:

The definition if the term “ENS genes” should be made at the first usage of the term online 170, rather than where it is later defined on line 207.

S1. Correct spelling of undetermined

S8. Column 1 header is missing the “t” in patient

Text in table description should include apostrophe to be patient’s

S9. Capitalize “Primer”

**Have all data underlying the figures and results presented in the manuscript been provided?**

Reviewer #1: **No: **As outlined in comments to the Authors:

1) There is no data table corresponding to the “Evaluation of candidate gene expression” to show the complete list of genes this analysis identified as expressed in the mouse ENS or the human embryonic colon.

2) A table should be provided to show the orthologous Mouse and Zebrafish genes for the 1216 Human genes or transcripts present in CNVs of group 1-3 and group 4, to include the supporting source reference for these orthologue assignments (i.e. HGNC, MGI, ZFIN, Ensembl). The table should also indicate if mouse or zebrafish orthologues of a human gene do not exist. Finally the table should indicate if mouse orthologues of a human gene are present or absent from the microarray datasets analyzed in this study.

Reviewer #2: Yes

PLOS authors have the option to publish the peer review history of their article (what does this mean?). If published, this will include your full peer review and any attached files.

Reviewer #1: No

Reviewer #2: No

---

## [Decision Letter · Decision Letter 2]

6 Jul 2021

Dear Dr Brosens,

We are pleased to inform you that your manuscript entitled "Size matters: large copy number losses in Hirschsprung disease patients reveal genes involved in enteric nervous system development." has been editorially accepted for publication in PLOS Genetics. Congratulations!

The revised manuscript was seen by reviewer #1 of the earlier submission; as you will see, they are positive and enthusiastic about the work.

Yours sincerely,

Gregory S. Barsh

Editor-in-Chief

PLOS Genetics

Gregory Copenhaver

Editor-in-Chief

PLOS Genetics

Comments from the reviewers (if applicable):

Reviewer's Responses to Questions

**Comments to the Authors:**

Reviewer #1: The changes made, including new experiments and substantial text changes, have significantly improved the logic and readability of this manuscript. This body of work nicely demonstrates how new technologies and cross-species strategies can be applied to human genetics studies.

**Have all data underlying the figures and results presented in the manuscript been provided?**

Reviewer #1: Yes

PLOS authors have the option to publish the peer review history of their article (what does this mean?). If published, this will include your full peer review and any attached files.

Reviewer #1: No

**Data Deposition**

http://datadryad.org/submit?journalID=pgenetics&manu=PGENETICS-D-20-01399R2

**Press Queries**

---

## [Editor Report · Acceptance letter]

4 Aug 2021

PGENETICS-D-20-01399R2 

Size matters: large copy number losses in Hirschsprung disease patients reveal genes involved in enteric nervous system development. 

Dear Dr Brosens, 

We are pleased to inform you that your manuscript entitled "Size matters: large copy number losses in Hirschsprung disease patients reveal genes involved in enteric nervous system development." has been formally accepted for publication in PLOS Genetics! Your manuscript is now with our production department and you will be notified of the publication date in due course.

With kind regards,

Andrea Szabo

PLOS Genetics

On behalf of:
